# Analyzing links between simulated Laptev Sea sea ice and atmospheric conditions over adjoining landmasses using causal-effect networks

Zoé Rehder[1,2,5], Anne Laura Niederdrenk[1], Lars Kaleschke[3,5], and Lars Kutzbach[4]

[1]Max Planck Institute for Meteorology, Bundesstraße 53, 20146 Hamburg, Germany
[2]International Max Planck Research School on Earth System Modelling, Bundesstraße 53, 20146 Hamburg, Germany
[3]Alfred Wegener Institute, Klußmannstr. 3d, 27570 Bremerhaven
[4]Universität Hamburg, Allende-Platz 2, 20146 Hamburg, Germany
[5]formerly at Universität Hamburg, Bundesstr. 53, 20146 Hamburg, Germany

**Correspondence:** Zoé Rehder (zoe.rehder@mpimet.mpg.de)

**Abstract.** We investigate how sea ice interacts with the atmosphere over adjacent landmasses in the Laptev Sea Region as a step towards a better understanding of the connection between sea ice and permafrost. We identify physical mechanisms as well as local and large-scale drivers of sea-ice cover with a focus on one region with highly variable sea-ice cover and high sea-ice productivity: the Laptev Sea Region. We analyze the output of a coupled ocean-sea ice-atmosphere-hydrological discharge model with two statistical methods. With the recently developed causal-effect networks we identify temporal links between different variables, while we use composites of high- and low-sea-ice-cover years to reveal spatial patterns and mean changes in variables.

We find that in the model local sea-ice cover is a driven rather than a driving variable. Springtime melt of sea ice in the Laptev Sea is mainly controlled by atmospheric large-scale circulation, mediated through meridional wind speed and ice export. During refreeze in fall thermodynamic variables and feedback mechanisms are important — sea-ice cover is interconnected with air temperature, thermal radiation and specific humidity. Though low sea-ice cover leads to an enhanced southward transport of heat and moisture throughout summer, links from sea-ice cover to the atmosphere over land are weak, and both sea ice in the Laptev Sea and the atmospheric conditions over the adjacent landmasses are mainly controlled by common external drivers.

## 1 Introduction — Laptev sea ice and permafrost

To better understand both the mechanisms behind as well as the strength of the interaction between sea ice and land we explore links between sea ice and the atmosphere over land and identify local and large-scale drivers of sea-ice cover in the Laptev Sea. Sea ice interacts with the atmosphere on different scales. However, while links from sea ice to large-scale atmospheric processes have been shown (e.g. Simmonds, 2015; Luo et al., 2017; Screen et al., 2018; Samarasinghe et al., 2019), the strongest coupling to the atmosphere is local (Screen and Simmonds, 2010; Screen et al., 2013). Sea ice influences near-surface temperatures by changing the local energy budget and regulating the moisture and energy which enter the lower atmosphere (Screen and Simmonds, 2010; Screen et al., 2013). This effect is more predominant in fall than in spring (Serreze

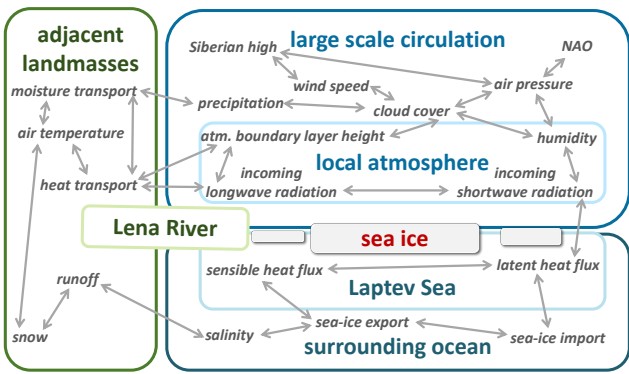

**Figure 1.** The regional climate system can be divided into general components (Laptev Sea, local atmosphere, adjacent land), which are embedded in the global climate system (surrounding ocean, atmospheric large-scale circulation) and interact with each other through a range of mechanisms including feedback loops; explanatory links indicated by grey arrows. All model variables used in the following analysis are listed in grey.

et al., 2009; Serreze and Barry, 2011; Screen et al., 2012). Additionally, downward radiation plays a role in changing the net surface fluxes and thereby the surface temperature. Downward radiation has been associated with moisture fluxes from mid-latitudes into the Arctic. The moisture fluxes show a positive trend in recent decades (Serreze and Barry, 2011; Lee et al., 2017).

Little attention has been focused on the physical mechanisms through which variability in sea ice influences the atmosphere over land. Nevertheless, from prior research we know that sea ice can exert such an influence on land (Lawrence et al., 2008; Ogi et al., 2016). Changes in the atmosphere over land which are attributed to declining sea ice lead to various responses in the permafrost landscapes, ranging from increased methane emissions (Parmentier et al., 2013, 2015) to changes in vegetation productivity (Bhatt et al., 2008; Macias-Fauria et al., 2017) and in vegetation composition (Post et al., 2013). Thus, a better

understanding of the connection between sea ice and land is valuable especially since sea ice and the permafrost covering adjacent landmasses are both highly vulnerable to climate change. In this paper, we aim for a better understanding of the physical mechanisms behind the connection of sea ice to the atmosphere over land.

As mentioned, sea ice has a strong impact on the energy balance of the ocean surface, giving rise to several feedback mechanisms, such as lapse-rate feedback (Pithan et al., 2013; Pithan and Mauritsen, 2014), water-vapour feedback (Francis

and Hunter, 2007) or ice-albedo feedback, which has been identified as a major control on Arctic temperatures (Deser et al., 2000; Serreze et al., 2009; Serreze and Barry, 2011; Graversen et al., 2014). These interconnections all contribute to a strong seasonality and interannual variability of sea ice, especially in the marginal seas of the Arctic Ocean (Deser et al., 2000). Because different processes might overlay each other when looking at the Arctic as a whole, we focus on one region where we expect a comparably strong connection of sea ice and the adjacent land: the Laptev Sea is one of the key contributors to net sea-

ice production in the Arctic (Bareiss and Görgen, 2005; Bauer et al., 2013) and shows large year-to-year variability (Haas and Eicken, 2001) (Fig. 2). To identify the main processes between sea ice and the atmosphere over permafrost, we include as many variables as possible in our analysis (for an overview of all included variables, see Fig. 1 and Tab. A1). Because observations

are sparse in space and time and not available for all relevant variables, we use model output. The big advantage of such a model over observational studies is that we can analyze a very large range of variables in a physically consistent system and in high resolution, both spatially and temporally. In contrast to reanalysis, we can run the model with the same forcing several times and can thus produce more data of stable climatic conditions. We are additionally able to compare different time scales and analyse the interactions on a monthly and daily scale. Previous studies have shown that unusual strong storm activities can change the state of Arctic sea ice in the long run (Screen et al., 2011; Simmonds and Rudeva, 2012, 2014). Such features on short time scales, for example the appearance of cyclones, can not be seen in an analysis based on monthly means only. However, for all model and reanalysis studies, it is important to keep in mind that the knowledge we can gain from looking at a larger scale is only as good as our understanding of the underlying process we depict in the model.

We use forcing from the time period of 1950 to 1989. In this period and in the Laptev Sea, we do not yet observe a general downward trend of sea ice. We run the model repetitively to improve statistical power. On the thus obtained 160 years of model output we employ two statistical methods. The first is called causal-effect networks, a recently developed method (Runge et al., 2012, 2014, 2015), which has been successfully applied by Kretschmer et al. (2016) to analyze Arctic drivers of mid-latitude winter circulation. This method allows us to a) identify the important links in an unbiased way and b) differentiate whether two variables are either subject to a common external forcing or which variable is forcing the other. Building on the results from the causal-effect networks, we group model years with exceptionally high and low sea-ice cover. These composites reveal spatial patterns and mean changes in variables, allowing us to gain a deeper physical understanding.

In the following, we start with introducing the causal-effect networks and the composite analysis. Then, we analyze the impact of sea-ice cover on the atmosphere over land as well as the drivers of sea-ice cover during the onset of the melting season and during refreeze. Finally, we put our results in the wider context.

## 2   Methods

In order to understand links between Laptev Sea sea ice and regional climatic conditions, we analyze the output of a regionally coupled ocean-sea ice-atmosphere-hydrological discharge model with two complementary methods. The model consists of the global ocean-sea ice model MPIOM (Jungclaus et al., 2013) coupled to the regional atmosphere model REMO (Jacob and Podzun, 1997; Jacob, 2001), which in our set-up covers most of the northern hemisphere. The atmosphere model has a horizontal resolution of approximately 55 km and 27 vertical levels. In the global ocean model, the grid poles are located over North America and Russia leading to a horizontal resolution of up to 5 km within the Arctic Ocean. The model has 40 vertical levels with varying depth. The regional atmosphere model and the ocean model in the uncoupled domain are forced with model output from the global model MPIOM/ECHAM5 for the time period 1950 - 1989 and the model was run repetitively. Details on the model set-up can be found in Sein et al. (2015) and Niederdrenk et al. (2013) and on the experimental design in Niederdrenk et al. (2016).

The model simulations have been validated against observations by Niederdrenk et al. (2013) and show a realistic mean Arctic climate for this time period. Also, the variability in sea-ice extent and thickness is captured well, being on the lower

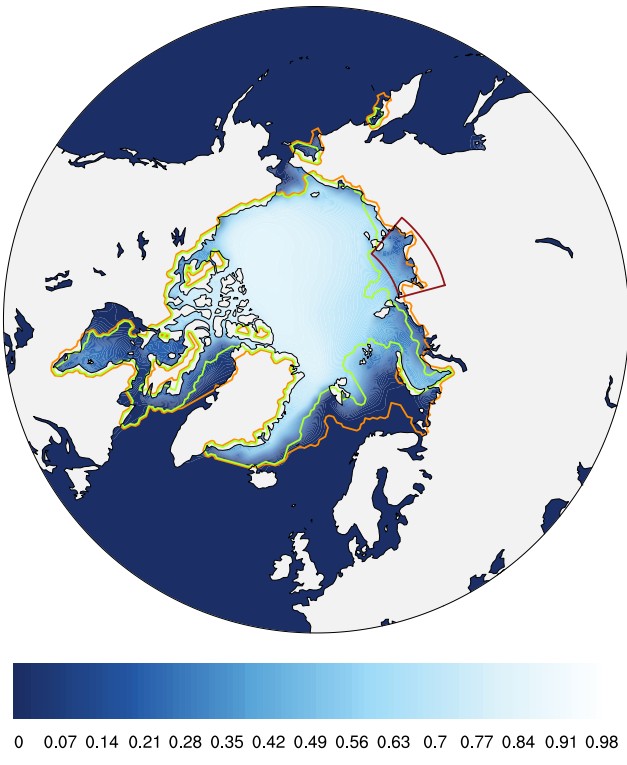

0  0.07  0.14  0.21  0.28  0.35  0.42  0.49  0.56  0.63  0.7  0.77  0.84  0.91  0.98

sea ice concentration [frac]

**Figure 2.** The July monthly-mean sea-ice concentration are taken from the model described in the main text. Red box indicates Laptev Sea Region and area over which one-dimensional time series of sea-ice cover are created. The contour lines show in green the minimum and in orange the maximum sea-ice cover. Note, that all of the ocean in the red box might be either ice-covered or ice-free and that the Laptev Sea is one of the areas in the model with the highest variability in monthly July sea-ice cover.

edge of observations. Due to its high resolution the regional model simulates more realistically than a global model the sea-ice transport within the Arctic (Niederdrenk et al., 2016). For our analysis, we run 40 years of forcing repetitively, so we can use 160 years of model output in total. Because the model melts ice directly from the ice edge, as it is not able to simulate land-fast ice and polynyas well, it shows less ice in the Laptev Sea than observations (see Fig.4). Nevertheless, the variability of the model lies within the observational records. For the chosen time period the output does not show a drift in sea-ice cover in the Laptev Sea (see Fig. 4) as sea-ice decline accelerated only in the nineties. Thus, we can use the model output to examine underlying links between the climate and Laptev Sea sea ice assuming no or only little disturbance through model drift or anthropogenic forcing. Sea-ice cover is the quantity we use to represent sea ice in our analysis since we expect that sea-ice cover has a larger impact on atmosphere-ocean exchange processes than e.g. sea-ice volume. Sea ice covers the Laptev Sea completely during winter, and most variability can be observed during the summer months (see Fig. 2). We therefore focus on the summer season to understand what influences sea-ice cover and what is influenced by sea-ice cover.

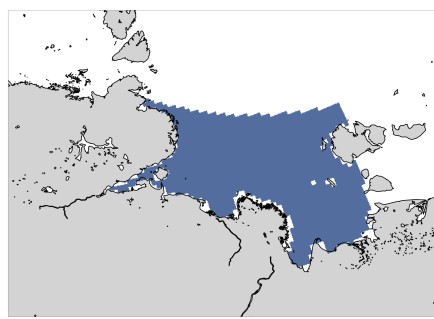

**REMO land and sea masks**   **MPIOM sea mask**

**Figure 3.** Colored areas show the model grid cells used in the analysis. Blue marks grid cells are used for ocean variables, and brown signifies land grid cells used to produce one-dimensional time series. MPIOM output is only available over water. REMO output was reduced either over land, over water or over both jointly. Colored areas lie between 105°E - 140°E and 70°N - 77°N.

As a first method we use causal-effect networks, implemented in the Python package TiGraMITe (Runge et al., 2012, 2014, 2015). Causal-effect networks is an algorithm for causal discovery: the algorithm finds causal links in a dataset without a-priori knowledge on physical mechanisms. Causal-effect networks determine how a perturbation moves through a set of one-dimensional time series. Each time series represents one variable, for example precipitation, and the temporal path of the perturbation is considered as a causal link. The procedure to find links and gauge their strength is divided into two steps. In the first, relevant causal and contemporary links for each time series are identified. In the second step, the strength of these links is quantified. To identify the links of a target time series, the correlation between this target time series and, one after the other, all other potentially driving time series are evaluated. For each variable, first the direct correlation is computed and then, in an iterative manner, the partial correlation by including all possible other time series. For each target time series, all potentially driving time series are also shifted back in time, as the signal in the driven time series will lag behind the signal in the driving time series. This shift is increased from one time step to a maximum time lag $\tau_m$. Only if a link remains significant no matter which subset of time series was included in the correlation analysis, we add a time series to the list of preliminary drivers of the target time series. This procedure is repeated for all time series in the dataset, so that we know all preliminary drivers for all time series. In a second step, these preliminary drivers are used to re-evaluate the link strength between each pair of time series by applying a multiple linear regression. We compute the multiple linear regression between a time series, the preliminary drivers of this time series, and, iteratively, one other time series at, one after the other, all possible time lags. The results of all multiple linear regressions are then summarized in a causal regression matrix of dimension $(N, N, \tau_m)$ where $N$ is the number of all time series we use and $\tau_m$ is the maximum time lag. Values in this matrix range from zero to one and are similar to a partial correlation coefficient. The threshold when a link is considered significant was set to $0.2$, such that in our networks links that are roughly two standard deviations stronger than the mean link between all time series are considered as significant. If we find a lagged link between two time series, we call it "causal". We also consider "contemporary" links, where the time lag is zero. For these links, it is not possible to determine the direction of information flow. Note, that the concept of

causation in causal-effect networks is related to Granger causality (Granger, 1969) which tests whether it is possible to predict the future development of one time series from the past development of another time series (Runge et al., 2014). Additionally, the first step of this algorithm is an adapted PC-algorithm (Spirtes et al., 2000) based on the idea that to identify causation we need to exclude common drivers (Pearl, 2000; Runge et al., 2012). Compared to many other frameworks making use of Granger causality the above algorithm is very computational efficient even on high dimensions (large $N \cdot \tau_m$) because using results of the first step drastically reduces the complexity of the second step (Runge et al., 2019).

The method becomes more reliable when the set of variables is large. To get a complete picture, we select all variables given in Fig.1 and Tab. A1. We spatially reduce the data by either computing averages or sums over the area of interest (red box in Fig. 2). If the variable has a "per-area"-unit, we weight the sum using the grid-cell area. Most of the variables are non-directional scalar fields, but vector fields, such as wind, we split into their zonal (positive sign: West to East) and meridional (positive sign: South to North) component and reduce the components separately. There are not only horizontal but also vertical fluxes. These we either integrate vertically over the atmosphere or only consider surface fluxes from atmosphere to land, ocean or ice. Lastly, for ice transport, we compute the gross import and export out of and into the Laptev Sea Region. The boxes for the atmosphere and ocean model are given in Fig. 3.

To resolve effects on different scales, we analyze both monthly and daily means. When analyzing monthly-mean time series, we consider the past year ($\tau_m = 12$ months) as maximal time lag, while we allow a lag of one week ($\tau_m = 7$ days) in the daily set-up. We look at the connection between land and sea ice especially during June - September when vegetation is photosynthesizing, and sea-ice cover is low and variable. This variability accentuates the differences between high and low sea-ice-cover years, which is important for the composite analysis. We additionally investigate what drives sea-ice evolution. To do so, we investigate the phases of strong changes in early ice melt and ice refreeze in spring and fall. While the Laptev Sea is still completely ice covered in March in most years, sea-ice cover starts declining in April in most years, so we consider the months April, May and June for ice melt. Starting in September, ice cover grows from its minimum extent to a nearly complete cover in November. Consequently, for the ice refreeze we focus on the months of September, October and November. Since our goal is to connect land and sea-ice cover, we only consider the atmosphere over landmasses adjacent to the Laptev Sea (see brown area in Fig. 3) for the summer period (June - September). In contrast, for early ice melt (April - June) and ice refreeze (September - November), we also want to understand what drives sea ice. So for these two cases, we look at the atmosphere over land and ocean together (entire shaded area in Fig. 3).

With causal-effect networks, we can disentangle temporal relationships. To also understand spatial patterns and quantify dependencies, we extend our analysis by using composites of years with exceptionally high and low summer sea-ice cover in the Laptev Sea between June and October. These months coincide with the months we chose for the summer causal-effect network, but are extended by October to include more of the refreezing phase, which only starts sometime during September. We select all years which deviate from the mean sea-ice state by more than 1.3 standard deviations. We select 1.3 standard deviations as a threshold to a) only use years with significant (larger than one standard deviation) anomalies in sea-ice cover; and b) to even out number of years contributing to the two composites as well as possible. With 1.3 standard deviations we get 17 years with high sea-ice cover and 7 with low, see Fig. 4.

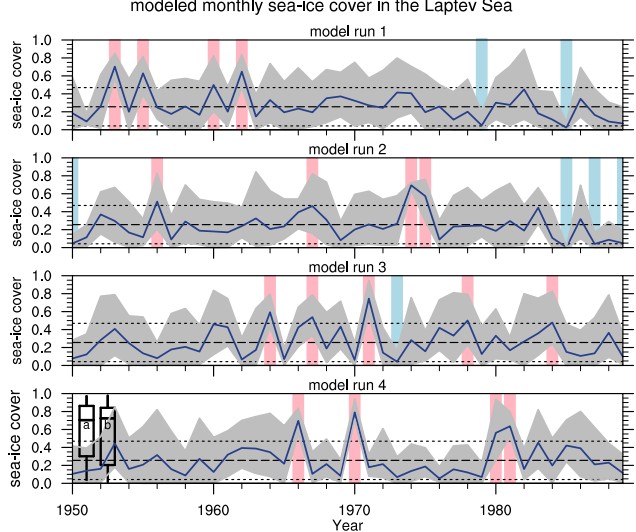

**Figure 4.** Monthly mean sea-ice cover in the Laptev Sea from June to October in each year. Blue line: mean sea-ice cover over summer and fall period. Grey area: range between minima and maxima of June to October monthly-mean cover. Middle dashed line: average sea-ice cover over all June-to-October means in all years and all model runs, upper/lower dashed line: threshold for composite members (mean $\pm$ 1.3 standard deviation), blue/red vertical lines mark composite members for low/high sea-ice cover. The boxplots in the lower panel show satellite data for comparison, using monthly mean ice cover in the Laptev Sea for the months June to October. The box indicates the first and third quartile, the middle line the median and the whiskers the minimum and maximum concentrations over all considered monthly means. Data for boxplot (a) is taken from the Hadley Centre (HadISST) for time period 1950-1990 (Titchner and Rayner, 2014). Data for boxplot (b) is taken from the National Snow and Ice Data Center (NSIDC) for time period 1979-1990 (Fetterer et al., 2016).

## 3 Results

As our main focus lies on the interaction between sea ice and land, we start our analysis by looking at the summer months when sea-ice variability is largest. Afterwards we will cover spring and fall to find drivers of the melt onset and the refreezing of ice.

### 3.1 Summer

To understand how sea-ice cover influences the adjacent land, we first look at causal-effect networks during the summer
150 months. Here, both on monthly and on daily scale, we only consider the atmosphere over land, not over the ocean. In this way, direct interactions between the sea surface and the atmosphere are excluded. In Fig. 5 (A), we see the results of the causal-effect networks analysis using monthly means throughout the summer months. Links with a strength below a threshold of $0.2$ are neglected as insignificant in our analysis. Out of the 16 variables used in the set-up, four are connected to sea-ice cover. The strength of the links to sea ice is weak (average strength: $0.25$): the (solely contemporary) connections the variables have between each other are much stronger (mean strength: $0.57$). For clarity, these connection are not shown in the figure. There is

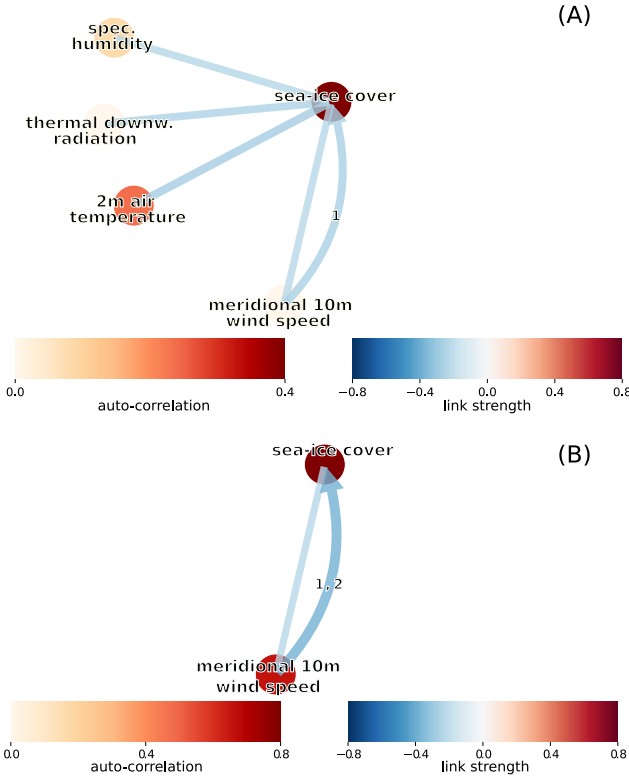

**Figure 5.** Causal-effect networks using atmospheric time series integrated over the land for summer based on monthly means (A) and daily means (B). The figure includes only those variables which have a significant connection to sea-ice cover. The color of nodes indicates auto-correlation. Straight lines show contemporaneous links, arrows show causal links with the time lag indexed on the arrow. The color of the arrows represents sign and strength of the link. Blue means that more of one variable coincides with less of the other, whereas red means that changes in the connected variables have the same sign. The link is stronger the darker the color. Variables with significant links to sea-ice cover are specific humidity (spec. humidity) averaged over vertical column, two-meter air temperature (2 m air temperature), meridional ten-meter wind speed (meridional 10 m wind speed) and thermal downward radiation at the surface (thermal downw. radiation).

**Table 1.** Minimal and maximal values for atmospheric meridional heat (Tv) and moisture (Qv) transport along a transect at 70N. Values were divided by the mean of each grid cell. Distributions of values for both variables displayed as boxplot in Fig. 6.

| sea ice | Tv | | Qv | |
|---|---|---|---|---|
| | min | max | min | max |
| low cover | -67.7 | 63.8 | -21.97 | 29.3 |
| mean cover | -90.6 | 90.4 | -31.7 | 58.3 |
| high cover | -87.5 | 69.0 | -25.6 | 37.7 |

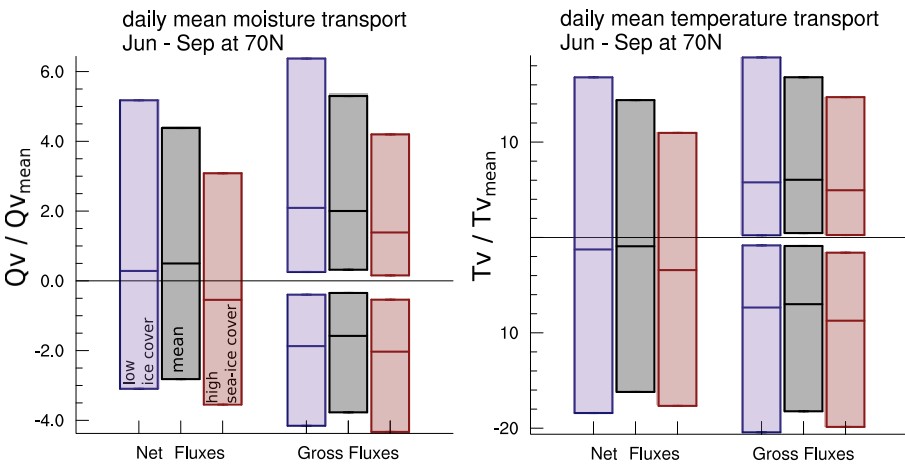

**Figure 6.** Meridional heat (Tv) and moisture (Qv) fluxes through a transect at the southern border of the area of interest (70 °N). Fluxes were divided by the mean flow through each grid cell over all years. In purple low-sea-ice-cover years are shown, in grey all years and in red high-sea-ice-cover years. 70°N lies just south of the coastline on land. We divide the fluxes into net (spatial mean over all data-points at the 70°N edge) and gross (spatial mean over all positive/negative data-points at 70°N boundary). Indicated are the median (centre bar) as well as the first and third quantile of the distribution. For minima and maxima, see Tab. 1.

only one causal link, from meridional wind speed to sea-ice cover, while all other links are contemporary. All these links have a negative sign: higher sea-ice cover is associated with a decrease in the strongly interconnected variables specific humidity and thermal downward radiation at the surface, as well as with 2 m air temperature and 10 m meridional wind speed. Sea-ice cover has fewer links on daily scale (see Fig. 5 (B)) than on the monthly scale. It only links with meridional wind speed, and
160 the only causal link is pointed towards, not away from, sea-ice cover meaning that sea-ice cover is the driven variable.

If we investigate wind with the composites by looking at mean wind speeds over the Laptev Sea Region, we see that wind direction is variable during the summer months, and there is a slight excess of southward wind. While in the mean about 56% of the days during June to September show winds pointing southward, about 53% of the days in the low-sea-ice-cover years and 64% of the days in the high-sea-ice-cover years have a mean southward wind direction: high sea-ice cover coincides with a
165 higher fraction of southward wind and vice versa for low sea-ice cover. More precisely, for high-sea-ice-cover years southward transport of moisture and heat is enhanced (see Fig. 6 and Tab. 1). The median of the net flux is lower compared to the mean and this southward shift even leads to a change in sign for moisture transport from net-northerly to net-southerly transport compared to the mean. For low-sea-ice-cover years, on the other hand, both north- and southward transport of moisture and heat increases. Again the median is lower than the median of all years for net fluxes. This indicates that the bulk of transport
also shifts towards more southward transport, but, for low sea-ice cover, there is a higher spread in the distribution visible in both the net as well as in the gross fluxes. Using a two-sample Kolmogorov-Smirnov test we find that for both moisture and heat transport the high-sea-ice-cover and low-sea-ice-cover distributions are significantly different (p-values < 0.001).

## 3.2 Early melt season of sea ice

Looking at monthly drivers of the early melt of sea ice in spring and including both the atmosphere over the ocean and over land, we find in total ten variables that have links to sea-ice cover (Fig. 7(A)). The gross of connections is contemporary (nine variables). Two causal links are found, both of them directed towards sea-ice cover and with a lag-time of one month. The lagged (causal) links are on average weaker than the contemporary ones. Returning to the contemporary links, the strongest links connect sea-ice cover to meridional wind speed, sea-ice export and meridional moisture transport, respectively.

On a daily scale (Fig. 7(B)), zonal wind speed and sea-ice export are still two of the three variables with the strongest connection to sea-ice cover. The third strongest connection now is with sea-ice import. These three directional variables also have causal links influencing sea-ice cover with a time lag of one and two days. Two other variables with causal links pointing towards sea-ice cover are the zonal vector components of wind speed and of moisture transport. Both have a one-day lag to sea-ice cover, and there is no contemporary link between sea-ice cover and either of the two. On a monthly scale, they have neither a contemporary nor a causal link. Links between sea-ice cover and sea-level pressure, surface longwave downward radiation and specific humidity on the other hand are significant on a monthly but not on a daily scale.

Since the strongest connections that sea-ice cover exhibits during early ice melt are connected to meridional transport and wind, we inspect spatial patterns using the composites for sea-level pressure in May (Fig. 8): we generally have a high over Greenland, but it is stronger for high sea-ice cover than for other years. For high sea-ice cover, we also observe a more distinct polar high, as opposed to low-sea-ice cover years, that are characterized by a strong high-pressure zone centered over the Chukchi Sea as well as a low over the Kara Sea. Locally over the Laptev Sea, these differences on the large scale lead to contrasting pressure fields. As air flows around pressure systems, we also have vastly different wind patterns in our area of interest. For high sea-ice cover, wind in the Laptev Sea predominantly blows from northeast, carrying air from similar or higher latitudes. For low sea-ice cover on the other hand, we have strong transpolar wind caused by the bipolar pressure field. Consequently, we also have a pronounced transpolar drift of sea ice originating roughly in the Laptev Sea and pointing towards Greenland. Over the Laptev Sea, the average wind direction is pointing northward for low sea-ice cover, and the air masses arrive from southeast, thus carrying comparably warmer air. In the monthly ice-melt causal-effect network (Fig. 7(A)), this warm air is reflected in a contemporary link between air temperature and sea-ice cover.

## 3.3 Refreezing of the sea in fall

The link between air temperature and sea-ice cover is also present during refreeze, when this is the strongest link of sea-ice cover to any variable (Fig. 7(C)) in the causal-effect network. As for spring, it was set up to include the atmosphere over both ocean and land. During ice refreeze, the links from sea-ice cover to sea-ice im- and export as well as meridional wind speed are weaker than during the melt onset. Heat and moisture transport do not have significant links to sea-ice cover. The autocorrelation of sea-ice cover, on the other hand, increased from 0.39 to 0.49. The auto-links of air temperature (strength: 0.40) and snow height (strength: 0.53) are also exceptionally strong compared to the other displayed variables. The total number of variables that are significantly connected to sea-ice cover decreases. The number drops from ten during ice melt to

eight during refreeze. Apart from meridional heat and moisture transport, the sensible surface heat flux and sea-level pressure disappear to be replaced by snow height and the latent surface heat flux. Both of these two variables have only causal and no contemporary links to sea-ice cover. The latent surface heat flux on the other hand is a dependent variable, with a time lag of three months. The lower sea-ice cover is, the higher are the absolute fluxes from ocean to atmosphere. The dependency of sea-ice export on sea-ice cover also changes: the contemporary and the causal link from sea-ice export to sea-ice cover during ice melt are replaced by a causal link from sea-ice cover to sea-ice export — more sea-ice cover leads to more export in the next month. In general, the number of causal links increases in the ice refreeze monthly-mean causal-effect network compared to melt. In addition, during refreeze sea-ice cover is also a preceding variable — during melt sea-ice cover is always the driven variable.

## 4  Discussion

### 4.1  Drivers of sea-ice cover

Tracing influences of sea-ice cover on the atmosphere over land with causal-effect networks is not straightforward: sea-ice cover couples more strongly to atmospheric variables integrated over both land and ocean (Fig. 7(A),(C)) than to the atmosphere over land (Fig. 5). For summer months, where we look solely at the atmosphere over land, we detect no causal link from sea-ice cover to any variable. The contemporary links that sea-ice cover has are weaker and smaller in number than in the causal-effect networks of melt and refreeze where we included the atmosphere over the ocean as well: sea ice has a stronger impact on the atmosphere directly over the ocean. This is in line with prior findings that the impact of sea ice on the atmosphere is mostly local (Screen et al., 2012, 2013).

A causal link we still observe is from meridional wind speed to sea-ice cover. Moreover, meridional wind speed is the only variable connected to sea-ice cover in the daily-mean summer causal-effect network, when only the atmosphere over land was considered. During early melt, when using both the atmosphere over land and over the ocean, we observe the same dependence of sea-ice cover on meridional wind speed.

In spring, we find a counter-intuitive link from air temperature to sea-ice cover (higher temperatures lead to more sea ice, Fig. 7(A)), which is most likely an artefact of the causal-effect network analysis because of our choice of time series: we include downward longwave radiation in our analysis, which is strongly connected to air temperature. But higher air temperatures coincide also with increased upward longwave radiation, which leads to an energy loss at the surface. This upward longwave radiation was not included in the causal-effect network. We think that within the analysis the positive link from temperature to sea ice is caused by an overcompensation: the additional energy available at the surface through downward radiation is subtracted from the influence of air temperature on ice but the energy radiated upward is not. This only becomes important for the causal link rather than the contemporary connection because for causal links we can subtract the influence of downward radiation at the present and prior time steps reinforcing the overcompensation.

Besides the link from air temperature to sea-ice cover, variables connected to the pressure field have the strongest links to sea-ice cover on both monthly and daily scale. Those include local sea-level pressure and particularly the north/south

components, like meridional wind speed, but also meridional moisture transport and sea-ice export. These variables are all strongly interconnected with each other on the monthly scale (Fig. 7(A)). While the variables have only contemporary links to sea-ice cover on monthly scale, they are driving sea-ice cover with a lag of one to two days in the daily-mean causal-effect network (Fig. 7(B)). This is in line with a prior observation-based study by Krumpen et al. (2013), in which sea-ice export from February to May has been singled out as one driver of sea-ice cover. The main local variable determining sea-ice cover in the Laptev Sea during early melt is thus meridional wind, which is also one main driver of sea-ice export. The stronger the northward flow, the less sea-ice cover. We can support this with our findings in the May composites (Fig. 8): for high-sea-ice-cover years, wind blows from the northeast carrying colder, less humid air, and, for low-sea-ice-cover years, winds are blowing from southeast enhancing transpolar wind and sea-ice export and bringing warmer, more humid air. (Meridional) wind depends on air pressure. The large-scale patterns of the May pressure field resemble the patterns of previous, mostly observational or reanalysis-based, studies (e.g. Wang et al., 2009; Jaiser et al., 2012; Overland et al., 2012). For high sea-ice cover, similar to previous results from Luo et al. (2019b), sea-level pressure patterns resemble the negative phase of the Arctic Oscillation index (AO) — a high over the central Arctic Ocean (Wang et al., 2009). We observe a pronounced high over Greenland. This might hint to Greenland blocking, an event which has been linked to a negative North Atlantic Oscillation index (NAO) (thus also a negative AO). Greenland blocking has been found to cause strong melt of the Greenland ice sheet (Overland et al., 2012) as well as an enhanced influx of moisture in the North Atlantic Region (Yang and Magnusdottir, 2017). This influx then enhances the water-vapour feedback. Instead of a positive AO pattern, low-sea-ice-cover years match an Arctic Dipole (AD) pattern (Fig. 8, similar to Wang et al. (2009); Overland et al. (2012)): we observe a high over the Chukchi and Beaufort Seas with even significantly higher pressure south of the Chukchi Sea and a low over the Barents Sea. The occurrence of the AD pattern fits well with the causal-effect-networks-based result, that meridional wind is strongly connected to sea-ice cover in the Laptev Sea. We have pronounced sea-ice export because the AD pattern leads to anomalies in the meridional wind and a stronger transpolar drift of sea ice, while the AO is more connected to zonal wind promoting either convergence or divergence of sea ice in the Arctic (Wang et al., 2009). Advection of Siberian air masses has been linked to extremely low sea-ice cover in the Laptev Sea before, for example by Haas and Eicken (2001). Apart from enhanced ice drift, the authors stress the importance of atmospheric heat content carried into the Laptev Sea. Additionally, temperature fluctuations caused by advection have been identified as a dominant driver of sea-ice variability by Olonscheck et al. (2019). This agrees well with our findings since, during melt, meridional wind, air temperature and meridional heat transport all have strong links to sea-ice cover. Contrasting, we find that the moisture content of the air transported into the Laptev Sea Region plays are a larger role than the heat content of the air. This is in agreement with a satellite- and reanalysis-based study by Yang and Magnusdottir (2017) who find that influxes of humid air into high latitudes from the North Atlantic lead to extremely low sea-ice cover in the Greenland, Barents and Kara Seas.

We conclude that during the melt phase, large-scale circulation systems exert a strong influence on the evolution of sea-ice cover. Since melting continues through summer (Kim et al., 2016) until September, the prevailing link from meridional wind speed to sea-ice cover portrays this connection.

During refreeze in fall, meridional wind speed is still important (Fig. 7(B)), but loses its dominance. While the connection between meridional wind speed and sea-ice cover in the monthly-mean causal-effect network for summer is on the same magnitude as the other connections of sea-ice cover, meridional wind speed has the weakest connection to sea ice among the significant variables during refreeze. Instead, state variables connected to feedback mechanisms, like ice-albedo feedback or water-vapour feedback, are increasingly important and also strongly interconnected. These feedback mechanisms we can already identify during early melt, even though they are less dominant then. In the fall monthly-mean causal-effect network (Fig. 7(C)), we observe a cluster of specific humidity, downward longwave surface radiation and air temperature. Specific humidity and downward longwave radiation are physically closely related as moist air reflects more longwave radiation than dry air, giving raise to the water-vapour feedback. Additionally, the auto-correlation of sea-ice cover and air temperature rises during refreeze. Since the strongest connection between air temperature and sea-ice cover is contemporary and the autocorrelation reaches back one time step, this can be interpreted as part of the strong interdependence of these variables. Sea-ice cover (increase) is also driving both latent surface heat flux and sea-ice export during refreeze in contrast to ice melt, when sensible heat fluxes and sea-ice export are driving sea ice (loss). The transition from mainly circulation- to mainly feedback-driven sea-ice cover is in agreement with Barton and Veron (2012) who found moisture-related feedback mechanisms to be a strong driver of Laptev Sea sea ice in September and October and in agreement with Arctic-wide findings that sea ice has a stronger impact on its surrounding in fall than in spring (Rigor et al., 2002; Serreze et al., 2009; Screen et al., 2013; Parmentier et al., 2015)

## 4.2 Influence of sea-ice cover on the atmosphere over land

What does this mean for the connection from sea-ice cover to the atmosphere over land? In summer, we observe a contemporary connection between sea-ice cover and specific humidity over land (Fig. 5). This link is also present in the ice-melt and refreeze monthly-mean causal-effect networks, where we consider the atmosphere over land and ocean in contrast to the summer, where we focus on the atmosphere over land. To better understand this connection, we inspect the composites. For high-sea-ice-cover years, we observe more days with on average southward wind — this is in compliance with our earlier findings: ice export is minimized, and wind tends to carry colder air into the Laptev Sea Region. More southward wind also means more southward transport of heat and moisture (Fig. 6 and Tab. 1). When open water areas are reduced compared to the mean state, both humidity and air temperature over the ocean are also reduced. So, the increase in gross southward transport of both moisture and heat can be attributed to the increase of air masses flowing south. At the same time, gross northward transport decreases. For low-sea-ice-cover years, there are fewer days with southward wind than usual. Still, we have a higher gross southward transport of heat and moisture because the air is warmer and moister over open water. More heat and moisture gets transported per air volume. In contrast to high-sea-ice-cover years, we also have an increase of transport northward corresponding to our earlier findings: low-sea-ice-cover years are caused by above average northward wind and an influx of warm and moist air from south. If we compute net moisture and heat fluxes for the low-sea-ice-cover years, the median is only marginally shifted towards an increase of southward transport for both variables. It is remarkable that the increase in southward transport more than balances the increase in northward transport we already identified above as the primary cause of above-average sea-ice cover decrease. We conclude that the contemporary links in the monthly-mean summer causal-effect network incorporate causality in both

directions: a decrease in sea-ice cover, as in the ice-melt period, is most likely initiated by increased meridional wind pointing north. But the increases in temperature and moisture over land, which then accompany low sea ice, are both cause, when transported northward, and result of the diminished ice cover due to consequently enhanced southward transport of heat and moisture in air. This is in line with a model study by Lawrence et al. (2008) who find that rapid sea-ice loss leads to warming on land. In contrast to their analysis, we break down this connection from sea ice to land and identify the responsible atmospheric variables. On the spatial (Laptev Sea) and temporal (days to months) scales analyzed here, we do not find a connection via large-scale circulation starting with changes in sea ice. However, we do not rule out that changes in whole-Arctic sea-ice cover or in other regions, such as the Kara and Barents Sea, can have an impact on land through large-scale circulation, such connections between sea ice and atmospheric circulation patterns have been shown before (e.g. Ogi et al., 2016; Yao et al., 2017; Luo et al., 2019a; Li et al., 2020). Concerning local and direct links from sea ice to land in the Laptev Sea Region, we find that over land the impact of sea ice is smaller than the impact of large-scale drivers, which simultaneously lead to changes in local sea-ice cover.

## 5  Conclusions

Concerning our research question on the drivers of the sea-ice variability, we conclude from our above model analysis:

- Drivers of sea-ice-cover variability differ during the early melt (April – June) and refreeze (September – November) period. During early melt, meridional wind and connected directional variables, like ice-export, are the prime drivers, all of them determined by the large-scale circulation. We find that high sea-ice cover often coincides with a negative Arctic Oscillation pattern, while low sea-ice cover coincides with an Arctic Dipole-like pressure pattern. During refreeze, in line with prior research (Rigor et al., 2002; Serreze et al., 2009; Screen et al., 2013), we conclude that thermodynamic feedback mechanisms related to temperature and moisture determine the speed of ice formation.

Regarding the connection between sea ice and the state of the atmosphere over land, we find:

- Sea-ice cover is primarily connected to the atmosphere directly above it and signals wash out quickly as we move to the land. Nevertheless, during low-sea-ice-cover years southward heat and especially moisture transport is enhanced. Heat and moisture are both variables which have a strong influence on the carbon budget as they affect both photosynthesis and respiration.

- We show that the impact of sea ice on moisture is at least as strong as on temperature. Studies which only focus on temperature overlook an important pathway of information, especially since the coupling is weak.

A possible explanation for the lack of causal links between atmosphere over land and sea-ice cover lies within temporal scales: the daily variations in the signal emitted by sea-ice cover have a significantly smaller impact than other forcings on the atmosphere over land. Otherwise, we would detect links on a daily scale. A potential slow but persistent forcing from sea-ice cover on the other hand might be diluted on a monthly scale due to the fast interactions within the atmosphere especially

because sea-ice cover exerts a remote forcing. We might still detect a link for the slow forcing, but most likely to several atmospheric variables and a contemporary instead of a causal link.

The above analysis was done using the climate of the last century before the onset of strong changes in the Arctic. The observed loss of sea ice in the Arctic as a whole has been projected to lead to changes in the general circulation pattern, namely via a shift to more negative Arctic Oscillation and North-Atlantic Oscillation indices, which favour high sea-ice cover and thus exert a negative feedback (Jaiser et al., 2012). No clear connection to anthropogenic forcing has yet been drawn for the Arctic Dipole, but from reanalysis we know that there is a higher persistence of this pattern in the 21st century compared to the last

(Overland and Wang, 2010). Depending on which of these two shifts persists over the other we can expect a faster or slower decrease of sea ice in the Laptev Sea.

Since sea ice is generally decreasing, we expect that southward heat and moisture transport from the open ocean to the land will increase. Higher temperatures lead to deeper thawing of the seasonally frozen soil layers during summer due to enhanced southward transport of heat and moisture in air. This increases soil-organic-matter decomposition in the then-unfrozen soils,

enhancing greenhouse gas emissions, but temperature is also one of the parameters steering the rate of photosynthesis, to name two of many processes which will be altered. Higher air moisture improves photosynthetic rates (high moisture leads to more efficient carbon fixation), changes the amount of (both short- and longwave) incoming solar radiation due to stronger reflection or, if higher humidity in the air leads to higher precipitation and water tables, the pathway of soil-organic-matter decomposition (Shaver et al., 2000). A general warming and an enhanced hydrological cycle are key features of global climate

change (Huntington, 2006; Stocker et al., 2013). In our model study we find that lower than usual sea ice in the Laptev Sea causes warming and an increase in air moisture over land, which might add to the above-mentioned trends. Nevertheless, we found the link from sea ice to land to be weak under stable conditions, and, if this relation holds under different conditions, we expect climate change over land to be driven primarily by large-scale circulation. Though a general decrease of sea ice in the whole Arctic has an effect on large-scale circulation (Jaiser et al., 2012; Li and Wang, 2013), which then effects the Arctic

landmasses, the local and direct link from sea-ice cover to adjacent landmasses appears marginal.

*Code and data availability.* Primary data and scripts used in this study are archived by the Max Planck Institute for Meteorology and can be obtained by contacting publications@mpimet.mpg.de. The model output is available through Niederdrenk and Mikolajewicz (2014) at https://cera-www.dkrz.de/WDCC/ui/cerasearch/entry?acronym=DKRZ_LTA_899_ds00003.

*Author contributions.* All authors designed the study together, ZR conducted the data analysis and prepared the manuscript, ALN provided

the model simulations and edited the manuscript.

*Competing interests.* The authors declare no competing interests.

*Acknowledgements.* We thank Thomas Kleinen and two anonymous reviewers for their helpful comments on the manuscript. We thank Marlene Kretschmer for her introduction to causal-effect networks. This work was partially funded by the Max Planck Society and by the DFG Cluster of Excellence 'CliSAP' (EXC177).

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

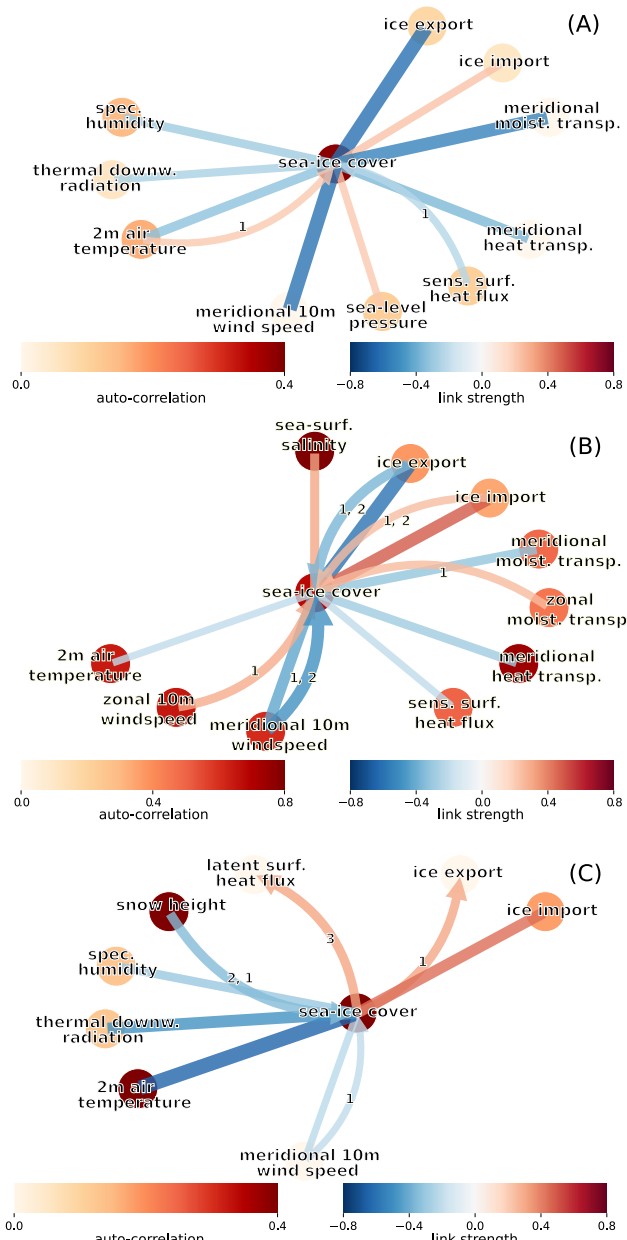

**Figure 7.** Causal-effect networks considering atmospheric variables over the whole Laptev Sea Region land and ocean area for ice melt and ice refreeze. (A) Ice-melt monthly mean. The connection in between the variables (not shown) have a mean strength of 0.44. Contemporary connections between surface longwave downward radiation, air temperature and specific humidity are especially strong (0.77). Same goes for sea-ice im- and export, meridional wind speed, heat and moisture transport (0.626). (B) Ice-melt daily means. Similar variables are dominant compared to (A), but more causal links are detected. Note, that the color scale for auto-correlation is different compared to (A) and (C). (C) Ice-refreeze monthly mean. Connections between variables (not shown) have a mean strength of 0.44. Contemporary connections between surface longwave downward radiation, air temperature and specific humidity are especially strong (0.79).

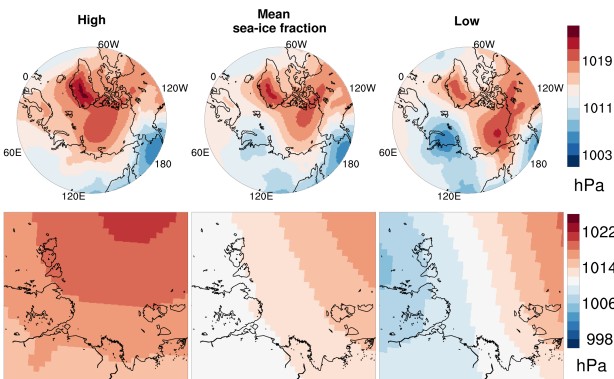

**Figure 8.** Sea-level air pressure in May on a large scale (upper row) and in the Laptev Sea (lower row). Centre column: May mean of all modelled years; Left/right column: mean over May mean in years of exceptionally high/low sea-ice cover in the Laptev Sea. Dots indicate areas which deviate by at least one standard deviation from the mean state (in the centre column).

| variable | reduction method | spring & fall | | summer | |
|---|---|:---:|:---:|:---:|:---:|
| | | mm | dm | mm | dm |
| fractional cloud cover | mean | ✓ | ✓ | ✓ | ✓ |
| fractional sea-ice cover | mean | ✓ | ✓ | ✓ | ✓ |
| gross mass export of sea-ice | transect sum* | ✓ | ✓ | | |
| gross mass import of sea-ice | transect sum* | ✓ | ✓ | | |
| meridional heat transport‡ | mean | ✓ | ✓ | ✓† | ✓† |
| zonal heat transport‡ | mean | ✓ | ✓ | | |
| height of ABL | mean | ✓ | ✓ | ✓ | ✓ |
| latent heat flux at surface | weighted sum | ✓ | ✓ | ✓ | ✓ |
| meridional moisture transport‡ | mean | ✓ | ✓ | ✓† | ✓† |
| zonal moisture transport‡ | mean | ✓ | ✓ | | |
| NAO index | pressure diff. | ✓ | | ✓ | |
| precipitation | sum | ✓ | ✓ | ✓ | ✓ |
| absolute downward LW radiation at surface | weighted sum | ✓ | ✓ | ✓ | ✓ |
| absolute downward SW radiation at surface | weighted sum | ✓ | ✓ | ✓ | ✓ |
| sea-level pressure | mean | ✓ | ✓ | ✓ | ✓ |
| sea-surface salinity | mean | ✓ | ✓ | | |
| sensible heat flux at surface | weighted sum | ✓ | ✓ | ✓ | ✓ |
| Siberian high index | regional SLP | ✓ | | ✓ | |
| snow height | mean | ✓ | ✓ | | |
| 2m temperature | mean | ✓ | ✓ | ✓ | ✓ |
| fresh-water flux from land to ocean | sum | ✓ | ✓ | ✓ | ✓ |
| spec. humidity‡ | mean | ✓ | ✓ | ✓ | ✓ |
| 10m meridional wind speed | mean | ✓ | ✓ | ✓ | ✓ |
| 10m zonal wind speed | mean | ✓ | ✓ | ✓ | ✓ |
| $\sum$ | | 24 | 22 | 18 | 16 |

**Table A1.** Time series included in causal-effect networks of monthly means (mm) and daily means (dm) to determine dominant drivers of sea-ice in spring and fall in the Laptev Sea as well as the influence of sea ice on the atmosphere over land during the summer. While atmospheric variables were integrated over both land and ocean for spring and fall, only the atmosphere over land was used in the summer causal-effect networks.

\* - sea-ice ex- and import are computed by summing the gross positive and negative values of transects at the outer borders of the areas indicated by the masks in Fig. 3.

† - to estimate the influence on land not the mean meridional transport was calculated but the flow through a transect at the southern border of the masked area.

‡ - vertically integrated over all atmospheric model layers.