# Peer review of "Analyzing links between simulated Laptev Sea sea ice and atmospheric conditions over adjoining landmasses using causal-effect networks"

_The Cryosphere, 2020_

## Referee Comment (RC1) · Anonymous Referee #1 · 30 May 2020

The study by Rehder et al. is an attempt to analyze links between sea ice dynamics in the Laptev Sea and the adjacent land. While the study is interesting, I have a couple of remarks about the model setup and its relevance in the context of earlier studies on this topic.

First of all, the authors find that the atmosphere mostly drives sea ice conditions in spring, that there's no strong link in summer between sea ice and the atmosphere (nor extending to the adjacent land), but that there's a stronger southward transport of both energy and moisture in low sea ice autumns, when the sea ice starts to freeze again. This is not a new finding. This has been shown before at the pan-Arctic level in several

publications by James Screen and co-authors (see e.g. Screen et al., 2012b, 2012a; Screen and Simmonds, 2010) but also others (For example Bintanja and Selten, 2014; Pithan and Mauritsen, 2014; Serreze et al., 2009; Serreze and Barry, 2011). It's surprising that none of these studies have been cited in this paper (although the authors cite another, less relevant, paper by Pithan et al. from the same year. Wrong citation perhaps?). At least some of these should be added next to the papers by Lawrence et al and Parmentier et al. that are already cited. Btw, the latter found strong correlations only in spring and autumn, but they argued that these correlations were contemporary in spring and only causal in the autumn, which corresponds to the findings by this study (but this is not mentioned here). The work by Graversen et al. is also a nice addition, since it shows a different view on the role of sea ice in arctic amplification (that northward atmospheric transport of heat may be more important). An alternate view on arctic amplification is given in the cited paper by Ogi et al but that's a very limited study of just nine weather stations, which is far from enough to grasp the drivers of arctic amplification beyond some local effects. While I appreciate the introduction of causal-effect networks to study ocean-atmosphere interactions, the general conclusions about the role of sea ice in ocean-atmosphere feedbacks are not new and the studied region is rather small, which makes it hard if not impossible to generalize to the whole of the Arctic.

Second, the paper starts of by presenting itself as a study where links are investigated between the ocean, the atmosphere and subsequently the land (i.e. permafrost thaw and carbon fluxes). However, despite using a regionally coupled model, they do not appear to have included a land surface model to actually model the response of the land surface (apart from runoff). So, in the end, the response of permafrost and carbon fluxes to changes in the atmospheric forcing due to sea ice decline remains unclear. The authors mention that this study is a first step, but the introduction suggests that this topic will be investigated in more detail – which isn't the case – and the topic doesn't come back until the conclusions as a possible outcome, but it has not been analyzed. So why lead with this topic in the first sentence of both the abstract and the main text
if the paper does not deal with this topic at all? Also here, the literature already holds many examples of possible connections which should be acknowledged if this topic is to be studied at a later stage (see e.g. Bhatt et al., 2010; Macias-Fauria et al., 2017; Parmentier et al., 2013; Post et al., 2013).

Apart from excluding a land surface model, the model setup also raises a few questions. First of all, why only focus on the Laptev Sea and the adjacent land? The regional model appears to have been run for most of the northern hemisphere and repeating the same analysis for other regions should be trivial. It would also show whether the found connections hold up in other regions where sea ice export is strong (e.g. along the coast of Greenland).

Also, why did the authors choose to run the model for the era before sea ice melt truly began (1950-1989)? This may lead to an underestimation of the role of sea ice in arctic climate feedbacks. If this is to be investigated, why not do this analysis for the period where sea ice started to decline and perhaps compare to the era of relatively stable ice conditions? The authors also repeat the same time period 4 times, but sea ice conditions are quite different between the four model runs. Why is this? It is not explained in the paper.

Overall, I think that the study is interesting, but the authors appear to present it as more novel than it is, and they should contextualize it better in the existing literature. A lot of work has been done on this topic, and a rather limited regional analysis over a historical time period with stable sea ice cannot be used in this way to draw strong conclusions on how sea ice decline has affected the whole arctic system, including the adjacent land, in recent decades.

A few other remarks:

- A diagram of which time periods and variables are compared to each other would be useful. From the text it can be difficult to follow which is being discussed. Perhaps label them?

[Figure]

- Page 5, line 94: which drivers of variables? Please specify.

- Page 10, line 195-196: why wasn't the causal effect network reanalyzed with long-wave radiation added? Seems important.

- Page 14, line 317-321: this conclusion is a rather big statement for an analysis of a limited area during an era of stable sea ice. It's not supported by this study nor the existing literature. Perhaps the link to land has been weak for the Laptev region during 1950-1989 but that doesn't mean it hasn't been strong in the past two decades in the same region or other parts of the Arctic!

References

Bhatt, U. S., Walker, D. A., Raynolds, M. K., Comiso, J. C., Epstein, H. E., Jia, G., Gens, R., Pinzon, J. E., Tucker, C. J., Tweedie, C. E. and Webber, P. J.: Circumpolar Arctic Tundra Vegetation Change Is Linked to Sea Ice Decline, Earth Interactions, 14(8), 1–20, doi:10.1175/2010EI315.1, 2010.

Bintanja, R. and Selten, F. M.: Future increases in Arctic precipitation linked to local evaporation and sea-ice retreat, Nature, 509(7501), 479–482, doi:10.1038/nature13259, 2014.

Macias-Fauria, M., Karlsen, S. R. and Forbes, B. C.: Disentangling the coupling between sea ice and tundra productivity in Svalbard, Scientific Reports, 7, 8586, doi:10.1038/s41598-017-06218-8, 2017.

Parmentier, F.-J. W., Christensen, T. R., Sørensen, L. L., Rysgaard, S., McGuire, A. D., Miller, P. A. and Walker, D. A.: The impact of lower sea-ice extent on Arctic greenhouse-gas exchange, Nature Climate Change, 3(3), 195–202, doi:10.1038/nclimate1784, 2013.

Pithan, F. and Mauritsen, T.: Arctic amplification dominated by temperature feedbacks in contemporary climate models, Nature Geoscience, 7(3), 181–184, doi:10.1038/ngeo2071, 2014.

Post, E., Bhatt, U. S., Bitz, C. M., Brodie, J. F., Fulton, T. L., Hebblewhite, M., Kerby, J., Kutz, S. J., Stirling, I. and Walker, D. A.: Ecological Consequences of Sea-Ice Decline, Science, 341(6145), 519–524, 2013.

Screen, J. A. and Simmonds, I.: The central role of diminishing sea ice in recent Arctic temperature amplification, Nature, 464(7293), 1334–1337, doi:10.1038/nature09051, 2010.

Screen, J. A., Deser, C. and Simmonds, I.: Local and remote controls on observed Arctic warming, Geophysical Research Letters, 39(10), L10709, doi:10.1029/2012GL051598, 2012a.

Screen, J. A., Simmonds, I., Deser, C. and Tomas, R.: The Atmospheric Response to Three Decades of Observed Arctic Sea Ice Loss, Journal of Climate, 26(4), 1230–1248, doi:10.1175/JCLI-D-12-00063.1, 2012b.

Serreze, M. C. and Barry, R. G.: Processes and impacts of Arctic amplification: A research synthesis, Global and Planetary Change, 77(1–2), 85–96, doi:10.1016/j.gloplacha.2011.03.004, 2011.

Serreze, M. C., Barrett, A. P., Stroeve, J. C., Kindig, D. N. and Holland, M. M.: The emergence of surface-based Arctic amplification, The Cryosphere, 3(1), 11–19, 2009.

---

## Referee Comment (RC2) · Anonymous Referee #2 · 10 Jul 2020

This submission examines links between sea ice variations in a small region of the Arctic and atmospheric structure over the neighboring landmasses to the south using a causal analysis approach. The specific Arctic region chosen is the Laptev Sea. The overall message contained in the results, consistent with many other recent studies (both model and observational), is that the sea ice in this broad longitudinal sector is to a great extent regulated by the largescale atmospheric circulation to the south and into the midlatitudes. The paper, for the most part, is well written and the methods and results appropriately presented. However, there are some areas where more explanation and clarity and a broader perspective are needed. The submission has the potential to make a significant contribution to the literature on this important and very timely topic,

but it is not quite there yet. Before I would be able to recommend acceptance, there are a number of issues which need to be addressed.

A significant proportion of the literature is quite old, and this must be updated. In my review I have pointed to relevant more recent studies in this rapidly evolving topic.

The investigation makes use of Jakob Runge's causal-effect networks approach. Methodologies of this general type are now in fairly common use in numerous fields (including climate analysis). Having said that, the reader who is not immediately familiar with the design and intricacies of such approaches would not be greatly helped by the brief description of the Runge algorithm presented here at Lines 86-99. The authors should present a broader and more informative qualitative and/or quantitative description of the procedure. This important step will make sure the reader fully appreciates the physical meaning of the results to come.

It would also be very helpful to comment on how the structure of Runge scheme compares with other packages which have been used in this sort of investigation. For example, Samarasinghe, S. M., M. C. McGraw, E. A. Barnes and I. Ebert-Uphoff, 2019: A study of links between the Arctic and the midlatitude jet stream using Granger and Pearl causality. Environmetrics, 30, e2540, doi: 10.1002/env.2540 used Granger causality and Pearl causality in their study of links between lower-latitude atmospheric circulation and Arctic temperatures. They stressed the value of comparing a range of different causality methods to fully understand the relevant processes. This important paper should be cited early in the manuscript as it deals with a topic closely related to the one examined here. Many readers may be familiar with the Granger causality paradigm ( C. W. J. Granger, 1969: Investigating causal relations by econometric models and cross-spectral methods. Econometrica, 37, 424-438, doi: 10.2307/1912791. C. W. J. Granger, 1980: Testing for causality: A personal viewpoint. Journal of Economic Dynamics and Control, 2, 329-352, doi: 10.1016/0165-1889(80)90069-x). A few words on how the structure of the present approach compares with that of Granger would be very helpful. (See also the paper of Marie C. McGraw, and Elizabeth A. Barnes, 2018:

Memory matters: A case for Granger causality in climate variability studies. Journal of Climate, 31, 3289-3300, doi: 10.1175/JCLI-D-17-0334.1.)

Lines 17-19: Here also cite the more recent analysis of Simmonds, 2015: Comparing and contrasting the behaviour of Arctic and Antarctic sea ice over the 35-year period 1979-2013. Ann. Glaciol., 56, 18-28.

Line 27-28: The reference relating to this important factor is very old. Update this by also referring here to analyses of Lee, S., S. B. Feldstein, . . ., 2017: Revisiting the cause of the 1989-2009 Arctic surface warming using the surface energy budget: Downward infrared radiation dominates the surface fluxes. Geophys. Res. Lett., 44, 10,654–10,661. Luo B., and co-authors (2017) Atmospheric circulation patterns which promote winter Arctic sea ice decline. Env. Res. Lett. 12, 054017, doi: 10.1088/1748-9326/aa69d0.

Lines 42-47: In this analysis the authors make use of the output of model simulations. They make the argument that they use model data 'to overcome limitations of observations, which are sparse in space and time and not available for all relevant variables'. I find this argument rather weak. Firstly there are a number of quality reanalysis data set which one could argue are the 'best' representation of the 4D atmospheric structure, which use all available observations plus the full gamut of (thermo)dynamic constraints implicit in the assimilating model. These do not extend over a 160-year period used here, but do go back many decades. Another critical issue on this point is that the model used (and indeed any model) will not capture all the appropriate physics, and hence the analysis of causality could be fraught. (Climate model shortcomings are particularly evident in, e.g., air-sea-ice interaction in the MIZ, capturing the Arctic Ocean horizontal and vertical structure, etc.). Overall, the use of model data in investigations such as this can be valuable, and I have no great problem with this here. However, think this justification comment should be presented more honestly, and presented alongside the range of caveats. The authors should remind the reader at appropriate intervals that the results are obtained using simulated data. Also, to make sure that the reader

is not misled the words 'simulated' or 'model' should appear in the title.

Lines 53-55: Related to the point of using model data, the authors have not really given us an idea of how realistically the means (and variability) are simulated. The only reference we are given for this is the mean SIC in July, with indications of the max and min (over the 160 years?) in Fig. 2. At least some comment should be made of how realistic this SIC simulation is. Also, the paper the sea ice melt and freeze periods – how well is the SIC simulated at those times of the year? The reader needs to know how well this basic parameter is represented. A worded comparison with the 'climatology' of Cavalieri, D. J., and C. L. Parkinson, 2012: Arctic sea ice variability and trends, 1979-2010. The Cryosphere, 6, 881-889, doi: 10.5194/tc-6-881-2012 should be presented.

Line 86: Python (sp.)

lines 132-133: A nice aspect of the paper is that it deals with both monthly and daily timescales. This dual perspective is a very important, as dealing only with the timescales beyond the synoptic time scale misrepresents the important associated with the presence of cyclones and fronts. These synoptic features exert great impact on the surface-atmosphere flues and sea-ice distribution. I strongly suggest this significant part of the analysis be emphasised here, and make reference to the following very relevant papers – Screen JA et al. (2011) Dramatic interannual changes of perennial Arctic sea ice linked to abnormal summer storm activity. J. Geophys. Res. 116: D15105 doi: 10.1029/2011JD015847 Simmonds et al. (2012) The Great Arctic Cyclone of August 2012. Geophys. Res. Lett. 39: L23709 doi: 10.1029/2012GL054259. Rudeva and co-authors (2014) A comparison of tracking methods for extreme cyclones in the Arctic basin. Tellus 66A: 25252 doi: 10.3402/tellusa.v66.25252.

Line 139-141: At some places in the text the authors make appropriate comments regarding the links when related, and confounding, processes are considered (e.g., at lines 207-213). This is a case in point here. The authors state that 'All these links have
a negative sign: higher sea-ice cover is associated with a decrease in specific humid-ity, thermal downward radiation . . .'. The ambiguity here is that the specific humidity is directly related physically to the thermal downward radiation, and hence these pa-rameters are not independent. On this connection valuable to reference the remarks of Screen, J. A. et al., 2018: Polar climate change as manifest in atmospheric circulation. Current Climate Change Reports, 4, 383-395, doi: 10.1007/s40641-018-0111-4 and a few words should be added as to how this can be disentangled.

Line 141: Change 'less links' to 'fewer links'

Lines 170-174: This observation in connection with the Greenland High etc. is con-sistent with the results shown by Luo et al., 2019: Weakened potential vorticity barrier linked to recent winter Arctic sea ice loss and midlatitude cold extremes. J. Climate, 32, 4235-4261. Beneficial to the argument to make reference to that paper here.

Lines 285-293: The authors remind us here that the study has considered the sea ice only in the Laptev Sea. It would be beneficial here (and elsewhere) to keep in mind that the nature of sea ice-atmospheric circulation relationships are known to depend strongly on the particular subregion of the Arctic that is being considered. In particular, sea ice in the Barents and Kara Seas to the west have considerable interactions with the Eurasian land mass of interest here, as distinct from the Laptev results presented in the paper. This is an important point to make, with some specific cases. In this context make reference to the recent studies of . . . Li, M . . .., 2020: Anchoring of atmospheric teleconnection patterns by Arctic sea ice loss and its link to winter cold anomalies in East Asia. Int. J. Climatol., doi: 10.1002/joc.6637. Luo, Wu, and . . ., 2019: The winter midlatitude-Arctic interaction: Effects of North Atlantic SST and high-latitude blocking on Arctic sea ice and Eurasian cooling. Climate Dyn., 52, 2981-3004. Yao and co-authors (2017) Increased quasi-stationarity and persistence of winter Ural Blocking and Eurasian extreme cold events in response to Arctic warming. Part I: Insights from observational analyses. J. Climate 30: 3549–3568

Lines 394-395: Please to present full details of this paper ... James E. Overland, Jennifer A. Francis, Edward Hanna and Muyin Wang, 2012: The recent shift in early summer Arctic atmospheric circulation. Geophysical Research Letters, 39, L19804, doi: 10.1029/2012GL053268.
* * *

---

## Author Response (AR1)

**Reply to the Editor**

We want to thank the Editor, David Schroeder, for his work. All suggested changes to the manuscript have been made.

This document comprises of the point-by-point replies to the comments of both reviewers as well as a marked-up version of the manuscript which includes the above-mentioned changes as well as the correction of a few more typos. We also updated the acknowledgements.

**Answer to RC1**

We thank the anonymous reviewer for their constructive feedback. Please find our answers below. The original review in italic, our answers in black font below and changes to the manuscript in bold.

1) First of all, the authors find that the atmosphere mostly drives sea ice conditions in spring, that there's no strong link in summer between sea ice and the atmosphere (nor extending to the adjacent land), but that there's a stronger southward transport of both energy and moisture in low sea ice autumns, when the sea ice starts to freeze again. This is not a new finding. This has been shown before at the pan-Arctic level in several publications by James Screen and co-authors (see e.g. Screen et al., 2012b, 2012a; Screen and Simmonds, 2010) but also others (For example Bintanja and Selten, 2014; Pithan and Mauritsen, 2014; Serreze et al., 2009; Serreze and Barry, 2011). It's surprising that none of these studies have been cited in this paper (although the authors cite another, less relevant, paper by Pithan et al. from the same year. Wrong citation perhaps?). At least some of these should be added next to the papers by Lawrence et. al and Parmentier et al. that are already cited. Btw, the latter found strong correlations only in spring and autumn, but they argued that these correlations were contemporary in spring and only causal in the autumn, which corresponds to the findings by this study (but this is not mentioned here). The work by Graversen et al. is also a nice addition, since it shows a different view on the role of sea ice in arctic amplification (that northward atmospheric transport of heat may be more important). An alternate view on arctic amplification is given in the cited paper by Ogi et al but that's a very limited study of just nine weather stations, which is far from enough to grasp the drivers of arctic amplification beyond some local effects. While I appreciate the introduction of causal-effect networks to study ocean-atmosphere interactions, the general conclusions about the role of sea ice in ocean-atmosphere feedbacks are not new and the studied region is rather small, which makes it hard if not impossible to generalize to the whole of the Arctic.

R1) These papers are valuable additions to our introduction. We included the mentioned papers in the first paragraph of the introduction, which we adapted as below. Additionally, we added selected papers as references at appropriate locations throughout the manuscript:

To better understand both the mechanisms behind as well as the strength of the interaction between sea ice and land we explore links between sea ice and the atmosphere over land and identify local and large-scale drivers of sea-ice cover in the Laptev Sea. Sea ice interacts with the atmosphere on different scales. However, while links from sea ice to large-scale atmospheric processes have been shown (e.g. Samarasinghe et al., 2019; Screen et al., 2018; Luo et al., 2017; Simmonds, 2015), the strongest coupling to the atmosphere is local (Screen and Simmonds, 2010; Screen et al., 2013). Sea ice influences near-surface temperatures by changing the local energy budget and regulating the moisture and energy which enter the lower atmosphere (Screen and Simmonds, 2010; Screen et al., 2013). This effect is more predominant in fall than in spring (Serreze et al., 2009; Serreze and Barry, 2011; Screen et al., 2012). Additionally, downward radiation plays a role in changing the surface fluxes and thereby the surface temperature. Downward radiation has been associated with the moisture fluxes from mid-latitudes into the Arctic, which show a positive trend in recent decades (Lee et al., 2017; Serreze and Barry, 2011). Little attention has been focused on the physical mechanisms through which variability in sea ice influences the atmosphere over land. Nevertheless, from prior research we know that sea ice can exert such an influence on land (Lawrence et al., 2008; Ogi et al., 2016). Changes in the atmosphere over land, which are attributed to declining sea ice, lead to various responses in the permafrost landscapes,

ranging from increased methane emissions (Parmentier et al., 2015, 2013) to vegetation productivity (Bhatt et al., 2008; Macias-Fauria et al., 2017) and vegetation composition (Post et al., 2013). Thus, a better understanding of the connection between sea ice and land is valuable, especially since sea ice and the permafrost covering adjacent landmasses are both highly vulnerable to climate change. In this paper, we aim for a better understanding of the physical mechanisms behind the connection of sea ice to the atmosphere over land.

Additionally, we added some of the papers also in the discussion to embed our findings better in the literature, like a reference to Parmentier et al. (2015) at the discussion of the fall fluxes. It is true that our study focus is on a very small region and we make our argument clearer for the choice of the Laptev Sea. Also, in our conclusions we only hypothesize what this could mean for the Arctic as a whole. Please also refer to our answer R9).

2) Second, the paper starts of by presenting itself as a study where links are investigated between the ocean, the atmosphere and subsequently the land (i.e. permafrost thaw and carbon fluxes). However, despite using a regionally coupled model, they do not appear to have included a land surface model to actually model the response of the land surface (apart from runoff). So, in the end, the response of permafrost and carbon fluxes to changes in the atmospheric forcing due to sea ice decline remains unclear. The authors mention that this study is a first step, but the introduction suggests that this topic will be investigated in more detail – which isn't the case – and the topic doesn't come back until the conclusions as a possible outcome, but it has not been analyzed. So why lead with this topic in the first sentence of both the abstract and the main text if the paper does not deal with this topic at all? Also here, the literature already holds many examples of possible connections which should be acknowledged if this topic is to be studied at a later stage (see e.g. Bhatt et al., 2010; Macias-Fauria et al., 2017; Parmentier et al., 2013; Post et al., 2013).

R2) We shifted the focus of the abstract by changing the first sentences as follows:

**We investigate how sea ice interacts with the atmosphere over adjacent landmasses in the Laptev Sea Region as a step towards a better understanding of the connection between sea ice and permafrost.**

All papers mentioned are now also included in the first paragraph of the introduction. See also R1).

3) Apart from excluding a land surface model, the model setup also raises a few questions. First of all, why only focus on the Laptev Sea and the adjacent land? The regional model appears to have been run for most of the northern hemisphere and repeating the same analysis for other regions should be trivial. It would also show whether the found connections hold up in other regions where sea ice export is strong (e.g. along the coast of Greenland).

R3) We want to look at physical mechanisms in depth, so we decided that it is more appropriate to focus on one region, rather than comparing several. We chose the Laptev Sea region, because it shows large interannual variability and borders on Eastern Siberia, which is covered by carbon-rich permafrost landscapes. The only other region with comparable interannual variability in the model is the Barents Sea, which is much more influenced by the North Atlantic than the Laptev Sea. Thus, for extracting the influence of sea ice on land, we deemed the Laptev Sea more fitting. Line 54:

The Laptev Sea is one of the key contributors to net sea-ice production in the Arctic (Bauer et al., 2013; Bareiss and Görgen, 2005) and shows large year-to-year variability (Haas and Eicken, 2001) as can be seen in Fig. 2. Its surrounding landmasses are characterized by near-pristine permafrost landscapes.

4) Also, why did the authors choose to run the model for the era before sea ice melt truly began (1950-1989)? This may lead to an underestimation of the role of sea ice in arctic climate feedbacks. If this is to be investigated, why not do this analysis for the period where sea ice started to decline and perhaps compare to the era of relatively stable ice conditions? The authors also repeat the same time period 4 times, but sea ice conditions are quite different between the four model runs. Why is this? It is not explained in the paper.

R4) Our aim is to first understand the underlying processes, before we investigate possibly interacting changes in the processes. Even if we might underestimate the effects of strong changes, we look at stable conditions instead to be able to extract the possibly weak signal from ice better. A possible next step would be to look at climate change.

The model has internal variability: The atmospheric model nearly covers the whole northern hemisphere and, consequently, can evolve freely without strong constrains by the external forcing. This is precisely the reason why we can run the model with the same forcing repeatedly, thereby prolonging the time series, without having the same values multiple times.

5) Overall, I think that the study is interesting, but the authors appear to present it as more novel than it is, and they should contextualize it better in the existing literature. A lot of work has been done on this topic, and a rather limited regional analysis over a historical time period with stable sea ice cannot be used in this way to draw strong conclusions on how sea ice decline has affected the whole arctic system, including the adjacent land, in recent decades.

R5) With the adjustments made in the manuscripts it should be clear, that we focus on the climate before warming and that we focus on one specific region.

6) A diagram of which time periods and variables are compared to each other would be useful. From the text it can be difficult to follow which is being discussed. Perhaps label them?

R6) We added a table providing an overview over the variables used in each set-up as well as, in the figure description, a summary of the analysis done. The table is appended to this document. This allows for a better overview. We added additional pointers throughout the paper as to which run was used for a certain conclusion.

**7) Page 5, line 94: which drivers of variables? Please specify.**

R7) To make it clearer, we changed the sentence as follows:

We look at the connection between land and sea ice especially during June - September when vegetation is photosynthesizing, and sea-ice cover is low and variable. This variability accentuates the differences between high and low sea-ice-cover years which is important for the composite analysis.

**8) Page 10, line 195-196: why wasn't the causal effect network reanalyzed with long-wave radiation added? Seems important.**

R8) Upward longwave radiation and temperature are highly correlated as the atmosphere is heated from below. To account for an influx of warm air (or cold air) we include the latitudinal and longitudinal temperature and moisture transport. Upward longwave radiation is also more directly connected to sea-ice cover than temperature. To reduce redundancies, we did not include upward longwave radiation in the analysis.

9) Page 14, line 317-321: this conclusion is a rather big statement for an analysis of a limited area during an era of stable sea ice. It's not supported by this study nor the existing literature. Perhaps the link to land has been weak for the Laptev region during 1950-1989 but that doesn't mean it hasn't been strong in the past two decades in the same region or other parts of the Arctic!

R9) With the changes below, the restrictions of the study are clearer.

A general warming and an enhanced hydrological cycle are key features of global climate change (Stocker et al., 2013; Huntington et al., 2006). In our model study we find that lower than usual sea ice in the Laptev Sea causes warming and an increase in air moisture over land, which might add to the above-mentioned trends. Nevertheless, we found the link from sea ice to land to be weak under stable conditions, and, if this relation holds under different conditions, we expect climate change over land to be driven primarily by large-scale circulation.

**Answer to RC2**

We thank the anonymous reviewers for their constructive feedback. Please find our answers to their comments below. The original review in italic, our answers in black font below and changes to the manuscript in bold.

1) The investigation makes use of Jakob Runge's causal-effect network's approach. Methodologies of this general type are now in fairly common use in numerous fields (including climate analysis). Having said that, the reader who is not immediately familiar with the design and intricacies of such approaches would not be greatly helped by the brief description of the Runge algorithm presented here at Lines 86-99. The authors should present a broader and more informative qualitative and/or quantitative description of the procedure. This important step will make sure the reader fully appreciates the physical meaning of the results to come.

R1) To start off with a more general introduction to the method we add the following sentences in line 72:

Causal-effect networks is an algorithm for causal discovery: the algorithm finds causal links in a dataset without a-priori knowledge on physical mechanisms.

Additionally, we expand more on the details of the description by adding the following information (new in bold):

The procedure to find links and gauge their strength is divided into two steps. In the first, relevant causal and contemporary links for each time series are identified. In the second step, the strength of these links is quantified.

To identify the links of a target time series, the correlation between this target time series and, one after the other, all other potentially driving time series are evaluated. For each variable, first the direct correlation is computed and then, in an iterative manner, the partial correlation by including all possible other time series. For all time-series, the time series are also shifted back in time, as the signal in the driven time series will lack behind the signal in the driving time series. This shift is increased from one time step to a maximum time lag taum. Only if a link remains significant no matter which subset of time series was included in the correlation analysis, we add another time series to the list of preliminary drivers of the target time series. This procedure is repeated for all time series in the dataset, so that we know all preliminary drivers for all time series.

In a second step, these preliminary drivers are used to re-evaluate the link strength between each pair of time series by applying multiple linear regression. We compute the multiple linear regression between a time series, the preliminary drivers of this time series, and, iteratively, one other time series at, one after the other, all possible time lags.

2) It would also be very helpful to comment on how the structure of Runge scheme compares with other packages which have been used in this sort of investigation. For example, Samarasinghe, S. M., M. C.

McGraw, E. A. Barnes and I. Ebert-Uphoff, 2019: A study of links between the Arctic and the midlatitude jet stream using Granger and Pearl causality. Environmetrics, 30, e2540, doi: 10.1002/env.2540 used Granger causality and Pearl causality in their study of links between lower-latitude atmospheric circulation and Arctic temperatures. They stressed the value of comparing a range of different causality methods to fully understand the relevant processes. This important paper should be cited early in the manuscript as it deals with a topic closely related to the one examined here. Many readers may be familiar with the Granger causality paradigm (C. W. J. Granger, 1969: Investigating causal relations by econometric models and cross-spectral methods. Econometrica, 37, 424-438, doi: 10.2307/1912791. C. W. J. Granger, 1980: Testing for causality: A personal viewpoint. Journal of Economic Dynamics and Control, 2, 329-352, doi: 10.1016/0165-1889(80)90069-x). A few words on how the structure of the present approach compares with that of Granger would be very helpful. (See also the paper of Marie C. McGraw, and Elizabeth A. Barnes, 2018: Memory matters: A case for Granger causality in climate variability studies. Journal of Climate, 31, 3289-3300, doi: 10.1175/JCLI-D-17-0334.1.)

R2) Thank you for your suggestion to include a comparison with a more common approach. We include a paragraph on Granger causality after line 84:

Note, that the notion of causation of the causal-effect networks is related to Granger causality (Granger, 1969) which tests whether it is possible to predict the future development of one time-series from the past development of another time series (Runge et al., 2014). Additionally, the first step of the causal-effect networks is an adapted PC-algorithm (Spirtes et al., 2000) based on the idea that we need to exclude common drivers to identify causation (Pearl et al., 2000; Runge et al., 2012).

Compared to many other frameworks making use of Granger causality causal-effect networks are very computational efficient even on high dimensions (large  $N \cdot tau_m$ ), because using results of the first step drastically reduces the complexity of the second step of the causal-effect networks (Runge et al., 2019).

Additionally, we cite the above-mentioned paper by Samarasinghe et al. (2019) at an appropriate place in the introduction.

*3) Lines 17-19: Here also cite the more recent analysis of Simmonds, 2015: Comparing and contrasting the behaviour of Arctic and Antarctic sea ice over the 35-year period 1979-2013. Ann. Glaciol., 56, 18-28.*

R3) It is a good idea to start the literature introduction with a study that deals with the connection of the atmosphere (via sea-level pressure) and sea ice in the Arctic as a whole. We change the lines as follows:

However, while links from sea ice to large-scale atmospheric processes have been shown (e.g. Samarasinghe et al., 2019; Screen et al., 2018; Luo et al., 2017; Simmonds, 2015), the strongest coupling to the atmosphere is local (Screen and Simmonds, 2010; Screen et al., 2013).

4) Line 27-28: The reference relating to this important factor is very old. Update this by also referring here to analyses of Lee, S., S. B. Feldstein, ..., 2017: Revisiting the cause of the 1989-2009 Arctic surface warming using the surface energy budget: Downward infrared radiation dominates the surface fluxes. Geophys. Res. Lett., 44, 10,654–10,661. Luo B., and co-authors (2017) Atmospheric circulation patterns which promote winter Arctic sea ice decline. Env. Res. Lett. 12, 054017, doi: 10.1088/1748-9326/aa69d0.

R4) We included both references within the introduction:

Sea ice interacts with the atmosphere on different scales. [...] Sea ice influences near-surface temperatures by changing the local energy budget and regulating the moisture and energy which enter the lower atmosphere (Screen and Simmonds, 2010; Screen et al., 2013). This effect is more predominant in all than in spring (Serreze et al., 2009; Serreze and Barry, 2011; Screen et al., 2012). Additionally, downward radiation plays a role in changing the surface fluxes and thereby the surface temperature. Downward radiation has been associated with the moisture fluxes from mid-latitudes into the Arctic, which show a positive trend in recent decades (Lee et al., 2017; Serreze and Barry, 2011).

5) Lines 42-47: In this analysis the authors make use of the output of model simulations. They make the argument that they use model data 'to overcome limitations of observations, which are sparse in space and time and not available for all relevant variables'. I find this argument rather weak. Firstly there are a number of quality reanalysis data set which one could argue are the 'best' representation of the 4D atmospheric structure, which use all available observations plus the full gamut of (thermo)dynamic constraints implicit in the assimilating model. These do not extend over a 160-year period used here, but do go back many decades. Another critical issue on this point is that the model used (and indeed any model) will not capture all the appropriate physics, and hence the analysis of causality could be fraught. (Climate model shortcomings are particularly evident in, e.g., air-sea-ice interaction in the MIZ, capturing the Arctic Ocean horizontal and vertical structure, etc.). Overall, the use of model data in investigations such as this can be valuable, and I have no great problem with this here. However, think this justification comment should be presented more honestly, and presented alongside the range of caveats. The authors should remind the reader at appropriate intervals that the results are obtained using simulated data. Also, to make sure that the reader is not misled the words 'simulated' or 'model' should appear in the title.

R5) To improve on the transparency concerning the use of models we will expand on the abovementioned paragraph as follows:

In contrast to reanalysis, we can run the model with the same forcing several times and can thus produce more data of stable climatic conditions. We are additionally able to compare different time scales and analyse the interactions on a monthly and daily time scale. [...] However, for all model and reanalysis studies, it is important to keep in mind that the knowledge we can gain from looking at larger scale is only as good as our understanding of the underlying process we depict in the model.

We do already include the word 'simulated' in the title and will add more reminders throughout the paper that we are using model data.

6) Lines 53-55: Related to the point of using model data, the authors have not really given us an idea of how realistically the means (and variability) are simulated. The only reference we are given for this is the mean SIC in July, with indications of the max and min (over the 160 years?) in Fig. 2. At least some comment should be made of how realistic this SIC simulation is. Also, the paper the sea ice melt and freeze periods – how well is the SIC simulated at those times of the year? The reader needs to know how well this basic parameter is represented. A worded comparison with the 'climatology' of Cavalieri, D. J., and C. L. Parkinson, 2012: Arctic sea ice variability and trends, 1979-2010. The Cryosphere, 6, 881-889, doi: 10.5194/tc-6-881-2012 should be presented.

R6) We make reference to a detailed comparison of the model results with observational records and reanalysis data within the text. Additionally, we now add observations into our figure and explain, why our model lies on the lower bound of the observations:

The model simulations have been validated against observations by (Niederdrenk et al., 2013) and show a realistic mean Arctic climate for this time period. Also, the variability in sea-ice extent and thickness is captured well, being on the lower edge of observations. Due to its high resolution the regional model simulates more realistically than a global model the sea-ice transport within the Arctic (Niederdrenk et al., 2016). For our analysis, we run 40 years repetitively, so we can use 160 years of model output in total. Because the model melts ice directly from the ice edge, as it is not able to simulate realistically land-fast ice and polynyas, it shows less ice within the Laptev Sea compared to observations (see Fig. 4). Nevertheless, the variability of the models lies within the observational records. For this time period the output does not show a drift in sea-ice cover in the Laptev Sea (see Fig. 4) as sea-ice decline accelerated only in the nineties.

**7) Line 86: Python (sp.)**

R7) Thank you! Spelling was corrected.

8) lines 132-133: A nice aspect of the paper is that it deals with both monthly and daily timescales. This dual perspective is a very important, as dealing only with the timescales beyond the synoptic time scale misrepresents the important associated with the presence of cyclones and fronts. These synoptic features exert great impact on the surface-atmosphere fluxes and sea-ice distribution. I strongly suggest this significant part of the analysis be emphasised here, and make reference to the following very relevant papers – Screen JA et al. (2011) Dramatic interannual changes of perennial Arctic sea ice linked to abnormal summer storm activity. J. Geophys. Res. 116: D15105 doi: 10.1029/2011JD015847 Simmonds et al. (2012) The Great Arctic Cy- clone of August 2012. Geophys. Res. Lett. 39: L23709 doi: 10.1029/2012GL054259. Rudeva and co-authors (2014) A comparison of tracking methods for extreme cyclones in the Arctic basin. Tellus 66A: 25252 doi: 10.3402/tellusa.v66.25252.

R8) We included the papers and highlight the usage of daily AND monthly data within our introduction (Line 42 and following):

We are additionally able to compare different time scales and analyse the interactions on a monthly and daily time scale. Previous studies showed that unusual strong storm activities can change the state of the Arctic sea ice in the long run (Screen et al., 2011; Simmonds and Rudeva, 2012, 2014). Such features on short time time scales, for example the appearance of cyclones, can not be seen in an analysis based on monthly means only.

9) Line 139-141: At some places in the text the authors make appropriate comments regarding the links when related, and confounding, processes are considered (e.g., at lines 207-213). This is a case in point here. The authors state that 'All these links have a negative sign: higher sea-ice cover is associated with a decrease in specific humidity, thermal downward radiation . . .'. The ambiguity here is that the specific humidity is directly related physically to the thermal downward radiation, and hence these parameters are not independent. On this connection valuable to reference the remarks of Screen, J. A. et al., 2018: Polar climate change as manifest in atmospheric circulation. Current Climate Change Reports, 4, 383-395, doi: 10.1007/s40641-018-0111-4 and a few words should be added as to how this can be disentangled.

R9) We add a remark that thermal downward radiation and specific humidity are connected at the above-mentioned location in the results section. Since we feel that an explanation would be better placed in the discussion section, we add the following sentences after line 242:

**Specific humidity and downward longwave radiation are physically closely related as moist air reflects more longwave radiation than dry air giving raise to the water-vapor feedback.**

The above-mentioned reference only briefly deals with moisture and longwave radiation, we add the reference at a more fitting spot in the introduction.

10) Line 141: Change 'less links' to 'fewer links'

R10) We corrected this.

11) Lines 170-174: This observation in connection with the Greenland High etc. is consistent with the results shown by Luo et al., 2019: Weakened potential vorticity barrier linked to recent winter Arctic sea ice loss and midlatitude cold extremes. J. Climate, 32, 4235-4261. Beneficial to the argument to make reference to that paper here.

R11) We include the above-mentioned paper in the discussion at a suitable place (Line 213 and following):

For high sea-ice cover, **similar to previous results from Luo et al. (2019b)**, sea-level pressure patterns resemble the negative phase of the Arctic Oscillation index (AO) – a high over the central Arctic Ocean (Wang et al., 2009). We observe a pronounced high over Greenland. This might hint to Greenland blocking, an event which has been linked to a negative North Atlantic Oscillation index (NAO) (thus also a negative AO).

12) Lines 285-293: The authors remind us here that the study has considered the sea ice only in the Laptev Sea. It would be beneficial here (and elsewhere) to keep in mind that the nature of sea ice-atmospheric circulation relationships are known to depend strongly on the particular subregion of the Arctic that is being considered. In particular, sea ice in the Barents and Kara Seas to the west have considerable interactions with the Eurasian land mass of interest here, as distinct from the Laptev results presented in the paper. This is an important point to make, with some specific cases. In this context make reference to the recent studies of . . . Li, M . . .., 2020: Anchoring of atmospheric teleconnection patterns by Arctic sea ice loss and its link to winter cold anomalies in East Asia. Int. J. Climatol., doi: 10.1002/joc.6637. Luo, Wu, and . . ., 2019: The winter midlatitude-Arctic interaction: Effects of North Atlantic SST and high-latitude blocking on Arctic sea ice and Eurasian cooling. Climate Dyn., 52, 2981-3004. Yao and co- authors (2017) Increased quasistationarity and persistence of winter Ural Blocking and Eurasian extreme cold events in response to Arctic warming. Part I: Insights from observational analyses. J. Climate 30: 3549–3568

R12) It makes sense to include comparisons to other regions, and we add the following description of the above-mentioned findings in line to clarify the scope of this study in comparison to other studies (after Line 272):

However, we do not rule out that changes in whole-Arctic sea-ice cover or in other regions, such as the Kara and Barents Sea, can have an impact on land through large-scale circulation, such as connections between sea ice and atmospheric circulation patterns have been shown before (e.g. Li et al., 2020; Luo et al., 2019a; Yao et al., 2017; Ogi et al., 2016).

13) Lines 394-395: Please to present full details of this paper ... James E. Overland, Jennifer A. Francis, Edward Hanna and Muyin Wang, 2012: The recent shift in early summer Arctic atmospheric circulation. Geophysical Research Letters, 39, L19804, doi: 10.1029/2012GL053268.

R13) We inserted the missing doi and numbers in the reference list:

Overland, J. E., Francis, J. A., Hanna, E., and Wang, M. Y.: The recent shift in early summer Arctic atmospheric circulation, Geophysical Research Letters, 39, L19 804, https://doi.org/10.1029/2012GL053268, 2012.

Additionally, we went through all references and included the DOI for all references.

**Analyzing links between simulated Laptev Sea sea ice and atmospheric conditions over adjoining landmasses using causal-effect networks**

Zoé Rehder1,2,5, Anne Laura Niederdrenk1, Lars Kaleschke3,5, and Lars Kutzbach4

1Max Planck Institute for Meteorology, Bundesstraße 53, 20146 Hamburg, Germany

2International Max Planck Research School on Earth System Modelling, Bundesstraße 53, 20146 Hamburg, Germany

3Alfred Wegener Institute, Klußmannstr. 3d, 27570 Bremerhaven

4Universität Hamburg, Allende-Platz 2, 20146 Hamburg, Germany

5formerly at Universität Hamburg, Bundesstr. 53, 20146 Hamburg, Germany

Correspondence: Zoé Rehder (zoe.rehder@mpimet.mpg.de)

Abstract. Studies based on satellite observations have shown that sea-ice cover and temperature in the Arctic tundra correlate well, and links between the two components have been suggested. We investigate links between sea ice and the state of the atmosphere over the adjacent landscapes with a focus on We investigate how sea ice interacts with the atmosphere over adjacent landscapes in the Laptev Sea , a marginal sea of the Arctic Ocean, which had highly variable summer-Region as a step towards.

- 5 a better understanding of the connection between sea ice and permafrost. We identify physical mechanisms as well as local and large-scale drivers of sea-ice cover with a focus on one region with highly variable sea-ice cover over the last century and high sea-ice productivity: the Laptev Sea region. We analyze the output of a coupled coupled a ocean-sea ice-atmospherehydrological discharge model under the climate of the last century utilizing composites of high and low sea-ice cover and a recently developed method called Causal Effect Networks, which identifies temporal links among one-dimensional time series.
- 10 Assuming that these temporal links indicate causation, we investigate how sea-ice cover may influence the atmosphere over land. We identify the main mechanisms driving ice melt in spring and refreezing in fallwith two statistical methods. With the recently-developed causal-effect networks we identify temporal links between different variables, while we use composites of high- and low-sea-ice-cover years to reveal spatial patterns and mean changes in variables.

In the model, ice melt We find that in the model local sea-ice cover is a driven rather than a driving variable. Springtime melt of sea ice in the Laptev Sea was is mainly controlled by atmospheric large-scale circulation, mediated through meridional wind speed and ice export. Thermodynamic During refreeze in fall thermodynamic variables and feedback mechanisms were important during refreeze are important - sea-ice cover was causally connected to is interconnected with air temperature, thermal radiation and specific humidity. Low ice cover led Though low sea-ice cover leads to an enhanced southward transport of heat , and moisture and low ice cover was linked to higher temperatures and higher humidity over land. However and moisture

20 throughout summer, links from sea-ice cover to the atmosphere over land were are weak, and both sea ice in the Laptev Sea and the adjacent landmasses were mainly driven externally. atmospheric conditions over the adjacent landmasses are mainly controlled by common external drivers.

Figure 1. The regional climate system divided into general components (Laptev Sea, local atmosphere, adjacent land), which are embedded in the global climate system (surrounding ocean, atmospheric large-scale circulation) and interact with each other through a range of mechanisms including feedback loops; explanatory links indicated by grey arrows. All model variables used in the following analysis listed in grey.

**1 Introduction - Laptev sea ice and permafrost**

- Attributed to global warming, recent years are filled with record lows of Arctic To better understand both the mechanisms
  behind as well as the strength of the interaction between sea ice and land we explore links between sea ice and the atmosphere over land and identify local and large-scale drivers of sea-ice cover , which expressively underpin a general downward trend (Dong et al., 2013). This trend is overlain by a strong seasonality and interannual variability, especially in the marginal seas (Deser et al., 2000), that cannot be explained by one main mechanism alonein the Laptev Sea. Sea ice strongly interacts with many components of the climate system, e. g. ocean salinity, air moisture and temperature fluxes, see Fig. 1 for more examples.
- 30 Notably, sea ice has a strong influence on the energy balance at the ocean surface. Less sea ice leads to more absorption of radiation, thus warming the surface and reducing ice further. This ice-albedo feedback has been identified as a major mechanism controlling Arctictemperatures and ice extent (Deser et al., 2000; Graversen et al., 2014). Apart from that, interacts with the thickness and state of the turbulently mixed atmospheric boundary layer controls whether air which was warmed at the ocean surface is quickly mixed upward and replaced at the surface or whether the warm air stays close to the surface.
- 35 The lower warm air is kept, atmosphere on different scales. However, while links from sea ice to large-scale atmospheric processes have been shown (e.g. Samarasinghe et al., 2019; Screen et al., 2018; Luo et al., 2017; Simmonds, 2015), the strongest coupling to the atmosphere is local (Screen and Simmonds, 2010; Screen et al., 2013). Sea ice influences near-surface temperatures by changing the local energy budget and regulating the moisture and energy which enter the lower atmosphere (Screen and Simmonds, 2010; Screen et al., 2013). This effect is more predominant in fall than in spring
- 40 (Serreze et al., 2009; Serreze and Barry, 2011; Screen et al., 2012). Additionally, downward radiation plays a role in changing the net surface fluxes and thereby the surface temperature. Downward radiation has been associated with the moisture fluxes from mid-latitudes into the Arctic, which show a positive trend in recent decades (Lee et al., 2017; Serreze and Barry, 2011)

. Little attention has been focused on the physical mechanisms through which variability in sea ice influences the atmosphere over land. Nevertheless, from prior research we know that sea ice can exert such an influence on land

45 (Lawrence et al., 2008; Ogi et al., 2016). Changes in the atmosphere over land, which are attributed to declining sea ice, lead to various responses in the permafrost landscapes, ranging from increased methane emissions (Parmentier et al., 2015, 2013) to vegetation productivity (Bhatt et al., 2008; Macias-Fauria et al., 2017) and vegetation composition (Post et al., 2013). Thus, a better understanding of the connection between sea ice and land is valuable, especially since sea ice and the more further warming of the atmosphere is enhanced (permafrost covering adjacent landmasses are both highly vulnerable to climate change.

50 In this paper, we aim for a better understanding of the physical mechanisms behind the connection of sea ice to the atmosphere over land.

As mentioned, sea ice has a strong impact on the energy balance of the ocean surface, giving rise to several feedback mechanisms, such as lapse-rate feedback) (Pithan and Mauritsen, 2014). The atmospheric boundary layer height is also one factor deciding how much moisture the air contains. Warm air can absorb more water, and, because water is a

- 55 strong greenhouse gas, moist air leads to more warming (water-vapor feedback ) (Francis and Hunter, 2007). The above feedbackmechanisms lead to local changes in energy fluxes, which in turn lead to changed fluxes within the atmosphere. This connection between sea ice and the atmosphere over the adjoining landmasses has been studied amongst others by Parmentier et al. (2015). In their analysis of satellite observations and several process-based methane models near-surface air temperature over land and sea-ice variability co-vary throughout many parts of the Arctic leading to the conclusion, that
- 60 local changes in the energy budget and ensuing temperature variations are the main assumed mediator of a causal link from sea-ice decline to increased methane emissions. Ogi et al. (2016) investigated links between observed sea-level air pressure, two-meter air temperature and sea-ice extent in Hudson Bay and feedback (Pithan and Mauritsen, 2014; Pithan et al., 2013) , water-vapour feedback (Francis and Hunter, 2007) or ice-albedo feedback, which has been identified as a major control\_on\_Arctic\_temperatures\_(Deser et al., 2000; Graversen et al., 2014; Serreze and Barry, 2011; Serreze et al., 2009).
- 65 These interconnections all contribute to a strong seasonality and interannual variability of sea ice, especially in the marginal seas of the Arctic Ocean - In contrast to Parmentier et al. (2015), they associated variability in both sea-ice cover and air temperature over the adjacent land with larger-scale atmospheric circulation fields. Additionally, Lawrence et al. (2008) found a connection between rapid sea-ice loss and temperatures on land, and Vaks et al. (2020) connected permafrost stability with the state (Deser et al., 2000). Because different processes might overlay each other when looking at the Arctic as a whole, we
- 70 focus on one region where we expect a comparably strong connection of sea ice during elimate states of the past. But neither of them explains, which changes in the atmosphere lead to and the adjacent land: The Laptev Sea is one of the link and what led to changes in the sea ice distribution.

In this study we want to complete this picture and disentangle causes and consequences of numerous climate variables. So far, there is no study including all possibly interacting variables pictured in key contributors to net sea-ice production in the

75 Arctic (Bauer et al., 2013; Bareiss and Görgen, 2005) and shows large year-to-year variability (Haas and Eicken, 2001) (Fig. 2). To identify the main processes between sea ice and the atmosphere over permafrost, we include as many variables as possible in our analysis (for an overview of all included variables, see Fig. 1, focusing on the question if and how variability

of Aretic sea ice influences the atmospheric state over the adjacent land masses. To overcome limitations of observations , which and Tab. A1). Because observations are sparse in space and time and not available for all relevant variables, we

- 80 utilize output of a regionally coupled climate model use model output. The big advantage of such a model over observational studies is that we can analyze a very large range of variables in a physically consistent system and in high resolution, both, spatially and temporally. To understand the fundamental interactions between land and sea ice, we use a pre-climate change time period of the last century and use in total 160 years of simulations, thus improving our statistical power. In contrast to reanalysis, we can run the model with the same forcing several times and can thus produce more data of stable climatic
- 85 conditions. We are additionally able to compare different time scales and analyse the interactions on a monthly and daily scale. Previous studies showed that unusual strong storm activities can change the state of the Arctic sea ice in the long run (Screen et al., 2011; Simmonds and Rudeva, 2012, 2014). Such features on short time time scales, for example the appearance of cyclones, can not be seen in an analysis based on monthly means only. However, for all model and reanalysis studies, it is important to keep in mind that the knowledge we can gain from looking at larger scale is only as good as our understanding of
- 90 the underlying process we depict in the model.

To find links within a set of variables and differentiate between spurious correlations, direct links, indirect links and contemporary neighbors we use We use forcing from the time period of 1950 to 1989. In this period and in the Laptev Sea, we do not yet observe a general downward trend of sea ice. We run the model repetitively to improve statistical power. On the thus obtained 160 years of model output we employ two statistical methods. The first is called causal-effect networks

- 95 (Runge et al., 2012, 2014, 2015). With this method, we gain understanding on temporal dependencies between different climate variables. Beside the analysis of the possible direct connection between sea ice and land temperatures we use the networks, a recently developed method (Runge et al., 2012, 2014, 2015), which has been successfully applied by Kretschmer et al. (2016) to analyze Arctic drivers of midlatitude winter circulation. This method allows us to a) identify the important links in an unbiased way and b) differentiate whether two variables are either subject to a common external forcing or which variable is
- 100 forcing the other. Building on the results from the causal-effect network to investigate the drivers of sea ice variability itself. To do so, we additionally compare extremely high and low networks, we group model-years with exceptionally high and low sea-ice extent events and the corresponding atmospheric states.

Due to the fact that different processes might overlay each other when looking at the Arctic as a whole, we rather focus on one region where we expect, if at all, a strong connection of sea ice and the adjacent land: The Laptev Seais one of the key

- 105 contributors to net sea-ice production in the Arctic (Bauer et al., 2013; Bareiss and Görgen, 2005) and shows large year-to-year variability (Haas and Eicken, 2001) as can be seen in Fig. 2. Its surrounding landmasses are characterized by near-pristine permafrost landscapes. The Permafrost region, where roughly 1300 Gt of soil organic carbon are stored (Hugelius et al., 2014), is , like sea ice, sensitive to climate change: With rising temperatures, the permafrost carbon pool is projected to be partly released to the atmosphere in form of carbon dioxide and methane (Schuur et al., 2015). Thus, we have two highly sensitive
   110 components of the climate system right next to each other, and so far it is unclear how they interact with each other.
  - In the following, focusing on the Laptev Sea region, we want to answer: Does sea-ice coverhave a causal impact on the atmosphere over land, and if yes, is temperature the main mediator as suggested before? What are the main drivers of local

**sea ice concentration [frac]**

Figure 2. July monthly-mean sea-ice concentration taken from the model described above. Red box indicates Laptev Sea region and area over which one-dimensional time series of sea-ice cover are created. The contour lines show in green the minimum and in orange the maximum sea ice cover. Note, that all of the ocean in the red box might be either ice-covered or ice-free and that the Laptev Sea is one of the areas in the model with the highest variability in monthly July sea-ice cover.

sea-ice variability?. These composites reveal spatial patterns and mean changes in variables, allowing us to gain a deeper physical understanding.

In this paperthe following, we start with introducing the causal effect-network-causal-effect networks and the composite analysis. Then, we analyze the impact of sea-ice cover on the atmosphere over land as well as the drivers of sea-ice cover during the onset of the melting season and during refreeze. Finally, we put our results in the wider contextof global climate change.

**2 Methods**

120 In order to understand links between Laptev sea ice and regional climatic conditions, we analyze the output of a regionally coupled ocean-sea ice-atmosphere-hydrological discharge model with two complementary methods. The model consists of